# Multistage Cloud-Service Matching and Optimization Based on Hierarchical Decomposition of Design Tasks

**Shuhui Ding** [1,2], **Zhongyuan Guo** [1], **Haixia Wang** [1,*] **and Fai Ma** [2]

1 College of Mechanical and Electronic Engineering, Shandong University of Science and Technology, Qingdao 266590, China
2 Department of Mechanical Engineering, University of California, Berkeley, CA 94709, USA
* Correspondence: wanghx@sdust.edu.cn

**Abstract:** In cloud manufacturing systems, the multi-granularity of service resource and design task models leads to the complexity of cloud service matching. In order to satisfy the preference of resource requesters for large-granularity service resources, we propose a multistage cloud-service matching strategy to solve the problem of matching tasks and resources with different granularity sizes. First, a multistage cloud-service matching framework is proposed, and the basic strategy of matching tasks with cloud services is planned. Then, the context-aware task-ontology modeling method is studied, and a context-related task-ontology model is established. Thirdly, a process-decomposition method of design tasks is studied, and the product development process with small granularity tasks is established. Fourthly, a matching strategy of ontology tasks and cloud services is studied, and the preliminary matching is accomplished. Finally, intelligent optimization is carried out, and the optimal cloud service composition is found with the optimal design period as the objective function. With the help of the preceding method, the service matching of maximizing the task granularity is realized on the premise of ensuring the matching success rate, which meets the preference of resource requesters for large-granularity service resources.

**Keywords:** cloud manufacturing; multistage cloud-service matching; design-task ontology modeling; semantic similarity matching; cloud service optimization; improved differential evolution algorithm

## 1. Introduction

By searching the cloud services provided by cloud providers in cloud manufacturing systems [1], the matching of design tasks posted by cloud requesters can be accomplished, which promotes the sharing of design resources and the networked implementation of design tasks. Design resources in cloud manufacturing systems [2] refer to all the elements involved in every design stage of the product life cycle, which are resource aggregations aggregated according to the size of function granularity. They are divided into dynamic capacity resource (DCR) and cross-functional design unit (CDU), which are used to complete feature-level design tasks and product-/system-level tasks, respectively [3]. At the same time, the design tasks posted by cloud requesters also exist in the form of different granularities of system tasks, subtasks or task units.

The multi-granularity of service resource and design task models leads to the complexity of cloud service matching. From the perspective of resource requesters, in order to facilitate the smooth design process and reduce the interaction with service providers in terms of parameters or design state, large-granularity service resources are preferred when the design capability, design cost and evaluation indexes are equal. Based on the multi-granularity of tasks and resources, as well as the preference of resource requesters for large-granularity service resources, we propose a multistage cloud service-matching strategy based on hierarchical decomposition of design tasks, which is used to solve the problem of matching between tasks and resources with different granularity.

The existing research similar to our study is mainly aimed at the design in non-cloud manufacturing environments, whose application environment is relatively simple. In the future cloud manufacturing environment, the design tasks proposed by resource users will be more complex and the composition of resources will be richer. Against such a background, this paper proposes to address the problems of decomposition of coarse granularity tasks and its multistage matching with cloud services. A multistage cloud service matching framework based on task hierarchical decomposition is presented. Under this framework, the context-aware task-ontology modeling method, information flow-based task decomposition, and semantic similarity-based matching strategy between ontology tasks and cloud services are studied. At last, based on an improved differential evolution algorithm, intelligent optimization is carried out for cloud services that meet the threshold requirements. The application of this research can realize the resource sharing in a cloud manufacturing environment and will provide convenient conditions for the integration and sharing of industrial design resources, which will build a bridge between highly specialized resources and interdisciplinary design tasks for resource owners and requesters.

The remainder of the paper is organized as follows. Section 2 reviews the literature relating to matching between tasks and cloud services, design process decomposition, and differential evolution algorithm-based intelligent optimization. In Section 3, a task hierarchical decomposition-based multistage cloud service matching framework is proposed. In Section 4, information flow-based task decomposition is created. The semantic similarity-based matching strategy between ontology task and cloud service is defined in Section 5. In Section 6, the intelligent optimization of cloud service is accomplished. The proposed methods are validated through a case study in Section 7, followed by the conclusion in Section 8.

## 2. Literature Review

In order to achieve the optimization matching between tasks and cloud services, research has been carried out in the fields of design task modeling and decomposition, matching tasks and cloud services, and optimizing cloud services.

### 2.1. Modeling and Decomposition of Design Task

In the aspect of design task modeling, most of the current literature only describes the task without specific modeling or does not make a standardized description of the user task. Due to the arbitrary description for the task, it is difficult to describe the user design intention completely and accurately. In the aspect of task decomposition, the commonly used method in the current literature is to transform tasks into standard service resource request descriptions, classify and extract all kinds of information, decompose I/O parameters and other basic information, and then directly match the corresponding information in the service resources to complete the service search and matching [4–6], or directly use the service resource words to search the cloud database [7].

The design tasks proposed by resource requesters [8] are often function-oriented. However, the functions of products or systems vary greatly according to their characteristics. They may be complex systems containing multi-disciplinary tasks, or the design of some parts or product parameters. In the matching process of task and cloud service, when the current cloud service or its combination cannot match the task directly, it is necessary to decompose the task to reduce the task granularity and improve the matching success rate. Aiming at the problems of matching between services and manufacturing tasks, Li et al. [9] proposed a novel service composition-optimization approach called improved genetic algorithm based on entropy. In this method, complex manufacturing tasks are decomposed into subtasks, which are accomplished by appropriate cloud services combined. Guo et al. [10] proposed an optimization mode of complex part machining services based on feature decomposition. In this method, the features of complex parts constitute basic task granularity. Guo [11] divided machining tasks into process level and part level and established a multi-granularity manufacturing-task model oriented to the machining field. Process-level

tasks have basic information, processing information, limited information and constraint information. Part-level tasks consist of basic information, parts processing information, service parameter information and constraint information. This method decomposes the processing tasks to a certain extent, but it does not specify the aggregation relationship between the two-level granularity tasks, so it is difficult to support the multi-granularity matching of later tasks and resources. The relative task-decomposition methods are shown as Figure 1.

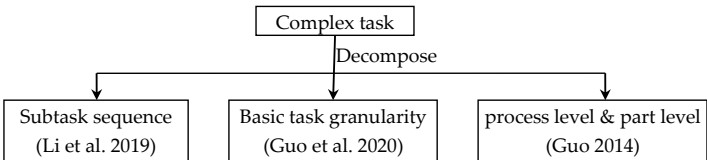

**Figure 1.** The relative task-decomposition methods.

In order to response to the task orders arriving in the cloud platform at any time, Zhang et al. [12] proposed a task-driven proactive service-discovery method, as well as a detailed description of task order, which contains the basic information and the functional requirement information. This task description can express the basic and functional related specifications but fails to express the task dynamic information. Aiming at the complexity of task decomposition, Yang et al. [13] introduced the graph theory to analyze the dependency relationship among tasks, proposed a task decomposition and allocation model based on hierarchical function—structure—task matching and collaborative partner fuzzy search, and established a design resource allocation model considering the resource utilization threshold. Deng [14] established a task-decomposition model based on cloud service primary selection in CMfg platform. Through the analysis of task cohesion and coupling, it ensured that the decomposition process met the principle of task competition. By analyzing whether the cloud service corresponding to the subtask was competitive, the phenomenon of inconsistency between task decomposition and service primary selection was solved. Liu et al. [15] designed a method of task decomposition and reorganization with adjustable granularity. Product was decomposed in detail according to the function, and the structure tree and corresponding fine granularity task set were established. Interval number was used to describe the information exchange degree between tasks to construct interval number DSM. Then, according to the required granularity, the parameter of possibility level λ was selected to establish the cut-off matrix. Finally, the tasks are clustered and reorganized to establish the final decomposition set of tasks. Yang et al. [16] decompose tasks into three categories: product requirements, expansion requirements and atomic requirements, and gave the multi-group formula of design requirements. This method of task definition failed to express the functional requirements and quality requirements of tasks, and the task description was incomplete.

Ontology is a powerful formal representation method, and the model built by ontology modeling technology has better advantages in semantic matching [17,18]. Han et al. [19] used task ontology based on hierarchical taxonomy so as to achieve productivity enhancement, cost reduction and outcome improvement through recommendations based on intelligence and personalization depending on the worker's present situation or context of task in charge when assembly of automotive parts is conducted. Lu et al. [20] proposed a kind of ontology product data model called ManuService, which includes information of product design and manufacturing, such as product specifications, quality constraints, and manufacturing processes. ManuService provides an approach to product design and manufacturing. Talhi et al. [21] presented an ontology-based model of the CMfg domain, where users can request services ranging from product design, manufacturing and all other stages of PLM. Whereas this ontology-based CM model is mainly for service resources, it is difficult to express tasks. Wang et al. [22] proposed a modeling framework of CMfg task to solve the General CMfg Task Ontology (GCMT_Ontology) construction and the task sub-ontology matching from GCMT_Ontology. Original CMfg task ontology is built from

CMfg task description model, and an ontology learning approach is put forward to complete GCMT_Ontology with graph-based semantic similarity algorithm. This method can improve enterprise collaboration and interoperation effectively. Dong [23] and Li [24] established a common domain ontology. In the process of service matching, service requesters and providers use this common domain ontology to accurately express the information that services need or can provide, such as input, output, premise and result information.

### 2.2. Matching between Design Task and Cloud Service

The matching of design tasks and service resources is crucial in cloud manufacturing systems, and relevant scholars have carried much research on it.

In [11], a framework of machining service discovery based on a multi-agent system is built, and a multi-granularity manufacturing service discovery process is formulated according to the framework composition pattern. According to the different characteristics of process-level and part-level tasks, the matching strategies of different levels of manufacturing services are constructed. Li et al. [25] presented a kind of decision-making model of the two-sided matching considering a bidirectional projection with preference information for hesitant fuzzy, which is applied to solve the configuration problem for cloud manufacturing tasks and resources. This model can help decisionmakers handle the configuration problems and has an advantage over the other approaches. Li et al. [24] constructed a resource matching model and transformed the resource discovery and matching problem into the mapping between resource ontology and task ontology to solve the matching problem. An intelligent matching algorithm is designed. The matching process is divided into five steps: state, domain, function, service and comprehensive matching to calculate the similarity. The candidate resources are sorted according to the similarity and the manufacturing is optimized resources.

Yu et al. [26] proposed a multi-level aggregate service-planning (MASP) methodology, and the MASP service hierarchy is presented, which deals with the services of multi-granularity and meets the requirements of all relater service providers. However, this method cannot deal with the matching of tasks and services. From the perspective of manufacturing resource retrieval accuracy, Yuan [27] studied the retrieval conditions proposed by resource users and semantic matching of manufacturing resources on the basis of an ontology-based weighted concept network and proposed an independent-element similarity algorithm and concept network-structure similarity algorithm. The language-based method is used to extract, process and calculate the concept of retrieval conditions.

### 2.3. Matching Calculation and Semantic Similarity Comparation

The calculation of matching degree between task and service resources is the key to resource discovery and the premise of service resource optimization [28]. Paolucci [29] proposed the concept of semantic matching of Web services earlier, and Sycara [30], Ganesan [31] and Kiefer [32] established detailed ontology-concept similarity algorithms. Tao [33] classified resource description information into four categories: text, sentence, value and entity, and designed their similarity algorithms respectively. On this basis, the manufacturing grid resource service matching is divided into four steps: basic matching, IO matching, QoS matching and comprehensive matching, which realizes the search and matching operation of service resources.

Sheng et al. [34] established an intelligent searching engine of CMfg service based on Ontology Web Language for Service (OWL-S) and analyzed its matching degree quantization problems of the matching process between ontology concept parameters and constraint parameters. With the help of this method, rapid intelligent matching of cloud searching was realized. Aiming at the matching degree of manufacturing resources and tasks, Li et al. [35] proposed a multi factor comprehensive matching evaluation model of resources and tasks. Through the analysis of manufacturing time-matching degree, manufacturing-fineness matching, comprehensive matching and other processes, manufacturing resources are optimized, so as to effectively utilize and reasonably allocate cloud manufacturing resources.

In order to realize the on-demand mutual selection between service providers and service demanders, Zhao et al. [36] proposed a two-way service matching model based on QoS. Qualitative language evaluation is mapped to quantitative evaluation by using a cloud model, and satisfaction is calculated by using variable fuzzy theory. With satisfaction maximization as the optimization solution goal, an optimization mathematical model is established and optimized, and finally a matching service was established. Nie et al. [37] proposed an improved ant colony algorithm based on a task resource matching function and cost function, which improved service quality by matching degree function of task and resource, reduced load imbalance in the cloud computing system by cost function and made great improvement in reducing task execution cost and improving system load balance.

Zhao [38] proposed a strategy of task and resource matching in collaborative development, established a matching optimization model based on multi-objective optimization, and proposed an improved particle-swarm optimization algorithm based on simulated annealing to solve the model, which realized the efficient optimization of product development. Wei [39] established a semantic-retrieval model of manufacturing resources based on rules and similarity and used the similarity algorithm based on semantic-weighted distance to calculate the semantic similarity between resource ontologies. Jiao et al. [40] proposed a method of service discovery about CMfg based on OWL-S, with which the algorithm for semantic similarity measurement in ontology is improved. This method can effectively distinguish the extent of similarity between CMfg services resources and service requirements.

### 2.4. Differential Evolution Algorithm-Based Intelligent Optimization

Differential evolution algorithm is a swarm-intelligence optimization algorithm proposed by Storn et al. [41], which uses floating-point vector coding for random search in continuous space [42]. Many scholars have improved it and applied it in resource matching and scheduling. Zhang [43] proposed a cloud-computing resource-scheduling strategy based on an improved differential evolution algorithm with adaptive mutation probability. The crossover operation was used to select individuals of the population to perform a multi-point crossover operation. The mutation operation could automatically set the threshold according to the fitness value, which increased the diversity of the population and the global search ability. Chen et al. [44] proposed an adaptive differential evolution algorithm based on Newton cubic interpolation. The Newton cubic interpolation is used to search the local near the optimal individual, and the adaptive demonstration strategy is designed to evaluate whether the Newton cubic interpolation is used in the next generation to avoid a premature algorithm and improve the performance of DE algorithm. Fan et al. [45] proposed an adaptive differential evolution algorithm with partition evolution of control parameters and adaptive mutation strategy. The mutation strategy can be automatically adjusted with the evolution of population, and the control parameters evolve in its own partition, which can automatically adapt and find the optimal value. Meng et al. [46] proposed a combined mutation strategy. Taking full advantage of each mutation strategy for population diversity index, a new parameter-control method for mutation operator, crossover operator and population size is proposed, which can solve a lot of optimization problems. Zhu et al. [47] proposed a method of gene location based on the subtask module level, and proposed three operators: block mutation, block crossover and block selection. On this basis, the transportation-scheme selection strategy and fitness-function calculation method were designed to make it more suitable for the actual working conditions of an optimal combination of cloud manufacturing resources.

### 2.5. Discussion

Through the above literature review on task- and resource-matching research, it can be concluded that the current research on task- and resource-matching methods has less consideration of resource and task granularity and has not considered the preference of resource requesters for large-granularity resources. Although some scholars have considered

the granularity of tasks [11], they do not match them according to the order of granularity, so they cannot match multistage resources according to the order of task granularity from coarse to fine.

Design task modeling is the prerequisite for task decomposition and resource matching. Most existing works only describe and normalize a single granular task, but do not elaborate the aggregation relationship between tasks. Guo [11] divides machining tasks into operation level and part level and establishes a multi-granularity manufacturing task model facing the machining field. Operation-level tasks are composed of basic information, machining information, restriction information and constraint information, and part-level tasks are composed of basic information, part processing information, service parameter information and machining constraint information. Wei [39] proposed a five-tuple formal definition of manufacturing resource ontology. Tao [33] proposed a manufacturing-grid resource-modeling method. The preceding method can standardize the modeling of design tasks, but it does not distinguish the granularity of tasks and the aggregation relationship between tasks of different levels of granularity, and it is difficult to support the multi-granularity matching of tasks and resources in the later stage.

In terms of task decomposition, the existing method is to convert the task into a service resource request description, analyze its basic parameters, and then match them with the corresponding information in the service resource [4–6], or search the cloud database directly using the service resource entry [7]. Yang et al. [13] proposed a task-decomposition and -allocation model based on the combination of hierarchical function structure task matching and collaborative partner fuzzy search. Deng [14] established a task-decomposition model based on the primary selection of cloud services in the cloud manufacturing platform. Through whether the cloud services corresponding to the sub-tasks after decomposition are competitive, the phenomenon that the task decomposition is inconsistent with the primary selection of services is solved. Liu et al. [15] decomposes products according to functions and established a design task decomposition set. The preceding design task-decomposition method does not involve the information flow between tasks of different granularity levels, and it is difficult to manage the key parameters between design tasks of different granularity. In this work, a process-decomposition method of design tasks based on information flow will be proposed, and the information flow will be used as a bridge between design resources, which solves the problem of parameter transfer between tasks of different granularities.

In terms of the matching and optimization of tasks and resources, most of the current literature focuses on resource discovery, without considering the preference of resource users for large-scale service resources. For example, Guo [11] develops a multi-granularity manufacturing service discovery process, which can build matching strategies for different levels of manufacturing services according to different characteristics of process-level and part-level design tasks. Li et al. [24] sorts the candidate resources according to the degree of similarity and optimizes the manufacturing resources by calculating the similarity between tasks and resources. From the perspective of manufacturing resource retrieval accuracy, Yuan [27] studies the retrieval conditions proposed by resource users and the semantic matching of manufacturing resources on the basis of an ontology-based weighted concept network. In the preceding studies, the task's granularity is not considered. We will propose a multi-level cloud-service matching strategy based on layer-by-layer decomposition of design tasks to meet the preference of resource users for large-granularity service resources on the basis of ensuring the quality of resource matching.

After servitization, the aggregated resources form cloud services, which are included in the cloud service pool. Due to the different sizes of resource aggregation, the service resources in the cloud service pool also show the characteristics of multi-granularity. Similarly, the tasks proposed by resource requesters can be divided into undecomposable feature tasks, parts tasks, component tasks, subsystem tasks, and system overall tasks according to the size of their functions. It can be seen that the granularity of the service model and task model is quite different.

In order to meet the requirements of resource requesters, and based on the consideration of resource providers and the cloud service platform, the following issues should be considered when matching services and task models with different granularity sizes:

(1) From the perspective of resource requesters, customers expect the resource providers to provide resource services with the largest granularity. The task of the same-size granularity involves less designers, and each designer needs to complete more tasks. The increase of resource granularity reduces the number of interfaces between customers and resource providers and reduces the number of communication links and confirmation times with designers, which can improve design efficiency and reduce design delay and error caused by poor communication.

The extreme case of a coarse-granularity resource service is that the task of the resource requester is completed by the service provided by one resource provider. In this case, the customer only needs to provide design parameters and design requirements information once, and only needs to check the design result once when the transaction is completed. On the contrary, the extreme case of a fine-granularity resource service is that the task has been decomposed many times, and all the task units in the final task sequence are indecomposable meta tasks. In this case, resource requesters need to exchange design parameters and design results with each resource provider. Compared with the first case, the number of communications with designers is greatly increased, and the uncertainty also increases, which is not conducive to the smooth design process.

(2) From the perspective of a cloud service platform, the cloud services in the cloud service pool are the result of the servitization of resource aggregations. There are two kinds of service resources of capability-level service and unit-level service, which can directly complete the feature-level meta task and the part-level or component-level task. For larger granularity tasks, it is necessary to retrieve and match the historical service composition or establish a new service composition according to the task requirements, which will increase the system cost.

(3) From the matching process of tasks and resource services, reducing the granularity of tasks will reduce the design parameters and the complexity of service resources, which could lower the difficulty of matching process. Additionally, more service resources that meet the requirements will be found, and more alternative resources are provided.

The matching strategy is shown as Figure 2.

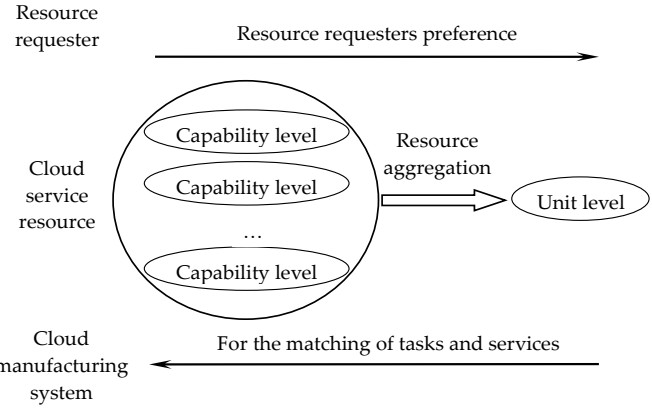

**Figure 2.** The matching strategy.

In order to meet the preceding matching principles of tasks and cloud services, this paper will study the overall matching strategy of tasks and resources, the modeling and decomposition of tasks, the matching of task ontology and cloud services, the intelligent optimization of cloud services, and the overall matching strategy of tasks and cloud services. This research will solve the optimization matching problem of the multi-granularity design

task model and service resources in cloud manufacturing systems under the premise of satisfying the preference of resource users for large-granularity service resources. Our method presented in this paper will facilitate resource users to find and match design resources in cloud manufacturing systems to meet their product design requirements.

## 3. Cloud Service Matching Strategy Based on Task Hierarchical Decomposition

According to the above strategy of matching tasks and services, the following matching rules are formulated:

Rule 1: task granularity maximization.

The coarse granularity service resources provided by a single resource provider can complete as many tasks as possible at one time. As it is completed by the same set of design resources, the interface and information interaction between multiple tasks occur within the design team, which is convenient for communication and can effectively reduce the information interaction between service providers and requesters, improve design efficiency and ensure design quality.

Rule 2: task hierarchical decomposition.

If the task proposed by the resource requester is an overall task, and cannot match any resource service successfully, the task will be decomposed into a design process layer by layer according to the sequence of subsystem task, component task, part task and feature-level meta task. Then, the design sequence is extracted, and each design sequence is matched with resources.

Rule 3: resource matching from fine to coarse, from simple to difficult.

When matching with a design task, firstly the cloud services in the cloud service pool are searched, and then the historical cloud service composition for the unsuccessful tasks. Finally, the service composition is conducted in the cloud service pool to establish the cloud service composition matching the task.

Based on the above matching rules, we propose a multistage cloud-service matching strategy based on task hierarchical decomposition. According to the matching requirements of tasks and semantic web services, a context-related task-ontology model is established.

### 3.1. Cloud-Service Matching Framework Based on Task Hierarchical Decomposition

Based on the preceding matching rules, and from the perspective of resource requesters, a multistage cloud-service matching method based on task hierarchical decomposition is established under the fundamental idea of "ensuring matching success rate and striving for maximum task granularity", as shown in Figure 3.

The construction process of cloud service matching is as follows.

(1)    Tasks published by resource requesters.

According to the requirements of the task-ontology model, resource requesters publish task instances. According to the task-ontology parsing algorithm, the cloud service platform extracts the service expectation information and task context information contained in the task ontology, such as function expectation, input and output parameters, service performance expectation, and the restrictions on service time, cost, etc.

(2)    First-level service matching.

The first level of service matching is to match the task with cloud services. The task information extracted in (1) is used to directly match the semantic similarity of a single cloud service in the cloud service pool. If successful, the matching is completed; otherwise, it enters the second-level matching.

(3)    Second-level service matching.

The second level is the matching of task and historical cloud service compositions. Historical cloud service compositions are cloud service compositions that have been successfully matched and successfully called in the early stage. As they have already performed

some tasks and have a certain maturity, reliable QoS information can be extracted, which will be used for the quick searching of a qualified mature resource team.

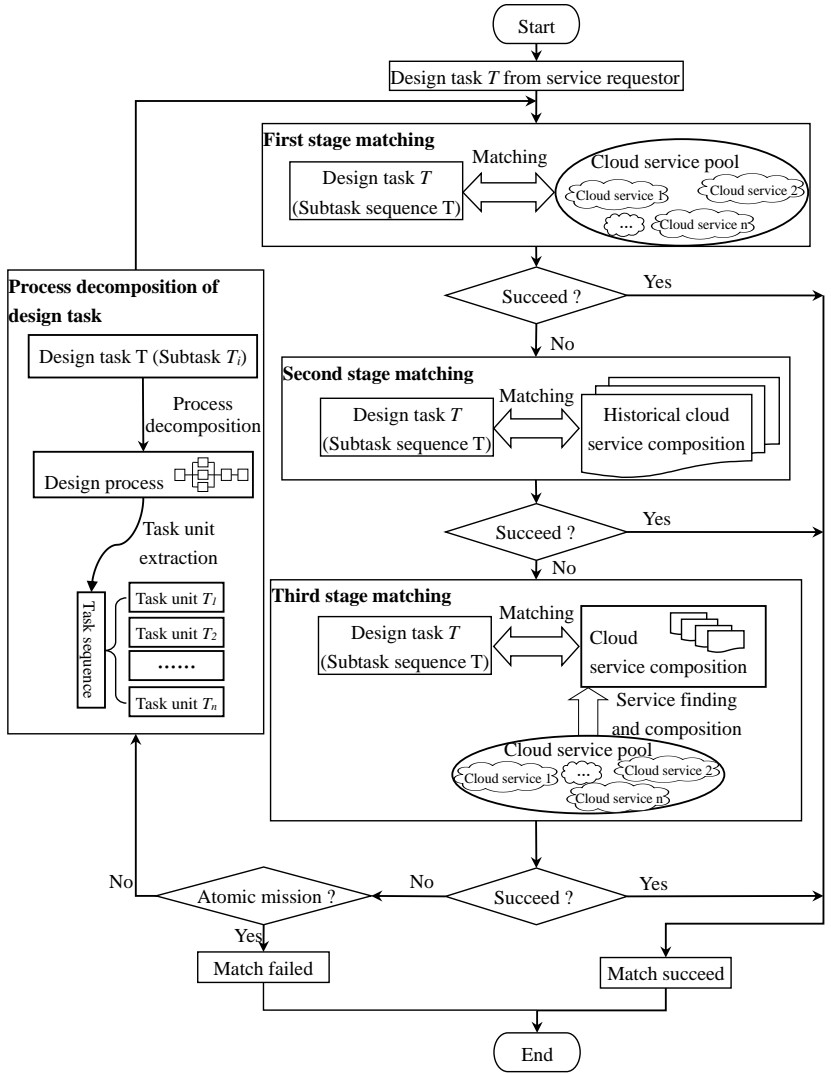

**Figure 3.** Multistage service matching based on task hierarchical decomposition.

(4)  Third-level service matching.

The third level of service matching is to find cloud services in the cloud service pool and build new cloud service compositions, and then match with the task. If the first two levels of matching fail, the service search and composition program will be invoked to search appropriate single cloud services from the cloud service pool and combine them into coarse granularity service compositions to complete the task.

(5)  Judgement of matching and meta task.

After the preceding three-level matching, if the task and resource match successfully, the service is invoked, and the task is accomplished. If the matching fails, judge whether the task is a meta task. If yes, it indicates that there is no relevant resource in the current cloud service pool that can meet the task requirements, and the matching fails and the program ends. If no, continue.

(6)  Task decomposition.

According to the sequence of system task, subsystem task, component task, part task and meta task, a task decomposition program is called to decompose the current-level task into the next-level design process, which is used to form a task unit sequence.

(7) For each task in the sequence of task units generated in (6), three-level service matching from (2) to (4) is performed. If it fails, the judgment procedure in (5) is executed, and then the further decomposition operation of (6) is executed and the next round of three-level matching operation is turned back, until all tasks match successfully or fail.

The service-matching method we proposed ensures the maximum task granularity on the premise of matching the task and cloud services as far as possible. This matching method starts with the original task with the maximum granularity as the starting point, aims at minimizing the running cost of CMfg system, and takes cloud service pool matching, historical cloud service composition matching and constructing cloud service composition matching as the order, and first carries out resource matching of maximum granularity tasks. If the matching fails, the task is decomposed to reduce the task granularity and improve the matching success rate of each subtask.

The pseudo code for matching of task and cloud service is shown as Algorithm 1.

---

**Algorithm 1.** The Algorithm of Matching between Task and Cloud Service

---

1. **Input:** design task $T$
    historical cloud service composition
    cloud services in cloud service pool
2. **Output:** the matching between task and cloud service
3. **define** task queue $TQ$
4. **define** task queue SC
5. **define** variable $i, j, k, h, m, n$
6. design task $T$ issued by service requester
7. $TQ \leftarrow T$
8. **while** (1)
9. $m \leftarrow$ the number of task in $TQ$
10. **for** ($i = 0; i < m; i ++$) **do**
11. matching between $TQ_i$ and cloud service accessed from cloud service pool
12. **end for**
13. **if** (task in $TQ$ matched) **then**
14. matching succeed and break
15. **end if**
16. **for** ($j = 0; j < m; j ++$) **do**
17. matching between $TQ_j$ and historical cloud service composition
18. **end for**
19. **if** (task in $TQ$ matched) **then**
20. matching succeed and break
21. **end if**
22. **for** ($k = 0; k < m; k ++$) **do** //third stage matching
23. $SC \leftarrow$ service composition from cloud service pool //construct service composition
24. $n \leftarrow$ the number of service composition in $SC$
25. **for** ($h = 0; h < n; h ++$) **do** //traverse $SC$ to search matched service composition with $TQ_k$
26. matching between $TQ_k$ and service composition $SC_h$
27. **end for**
28. **end for**
29. **if** (task in $TQ$ matched) **then**
30. matching succeed and break
31. **else**
32. **if** (each task in $TQ$ is atomic) **then** //judge task decomposable
33. match failed and break
34. **else** //decompose task and input to task sequence $TQ$
35. subtask sequence $\leftarrow$ process decomposition
36. $TQ \leftarrow$ subtask sequence
37. **end if**
38. **end if**
39. **end while**
40. **return**

---

### 3.2. General Matching Process between Task and Semantic Web Service

The matching process of task and cloud services is a typical application of a web service. Its basic idea is to search and match the web service description file to find the optimal web service that can accomplish the task and establish the service cooperation relationship with the design resource aggregation by calling the service. Using semantic web services, OWL-S ontology is used to formally define and describe web services when establishing their service description files, which improves the precision in the process of service searching.

The general matching process between tasks and semantic web services is shown in Figure 4, which is described as follows.

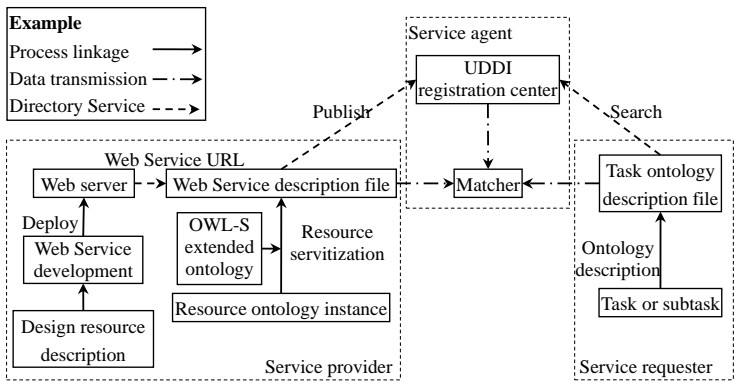

**Figure 4.** The universal matching process of ontology task and semantic web service.

(1)    Construction and publishment of semantic web service.

It includes two parts: the compilation of web service application and the generation and publication of semantic web service description file.

The web service application is written with programming language and tools such as Visual Studio.NET Programming environment using C# language, JDK programming tools using Java language, etc. Depending on the forms of web services, the functions in product design, such as calculation, query, etc., can be realized by programming, or the information such as design resources can be packaged to establish components and preset API so that the client can call web services through programming.

The semantic web-service description file is an OWL file generated by a semantic description of design resource information, with the help of semantic-description ontology framework established in [48]. The description files are stored in a web server, and the semantic web services are published by publishing its directory to Universal Description Discovery and Integration (UDDI) registration center.

(2)    The description and request of tasks.

The tasks are described as task ontology, from which the relevant information is extracted. The semantic description of a service request is transferred to UDDI to find related services.

(3)    Matching of service resources and tasks.

The matcher obtains the semantic description information of web services and tasks from the UDDI registration information, judges their matching degree, and returns the optimal result to the service requester.

In the matching strategy of this paper, the matching of tasks and service resources in each level of granularity goes through three stages: service pool matching, historical service composition matching and new service composition matching.

### 3.3. Context Aware Task Ontology Modeling

In the current research on the matching between tasks and resource services, most of the literature only describes the tasks without detailed modeling. There is no standardized description of tasks, which cannot fully and accurately express the user's design intention.

In the process of service matching, Dong et al. [23] used the same ontology model to construct tasks and resources models. However, from the perspective of service providers and consumers, service description is asymmetric, which shows that it has different contents and forms. For service providers, in addition to describing the name, classification, quality and other basic information of service providers, it is also necessary to describe the function information such as service input, service output, service preconditions and execution effect, basic service process information, service handover agreement, message format and other specific access details, so as to facilitate the search, optimization and access of services. For service consumers, it only needs to provide the basic information, the expected function information and the performance information of the task, as well as the consumers' contextual information, such as basic context information, static context information and dynamic context information. As mentioned above, when the service provider and the requester adopt the same ontology, they cannot accurately express the information of the service. Therefore, it is necessary to model the service ontology and the requirement ontology separately [21].

Based on the above research on the existing task-modeling methods, we have not found a service-matching method that can support the task decomposition proposed in this paper. Based on the preceding, a task-ontology modeling method based on service user context awareness is proposed to accomplish the accurate description of tasks and complete the optimal matching between tasks and resource services.

Context refers to the user's surrounding environment information, their own personalized information and the hardware environment information when they perform the task [49,50]. For the description of tasks proposed by resource users, the existing semantic-description methods basically do not consider the user context information [51], which plays an important role in the matching of resources and services and the optimization of matching results. In this paper, based on the requirements of an ontology model for user context and service expectations, the task-ontology modeling method establishes the formal definition of tasks by describing the service expectation of tasks and the ontology description of the user's own context information.

The content of task ontology includes service expectation and task context. The former is the expectation of task on the function, performance and fundamental service information of service resources, including the following aspects.

(1) Expectation of service fundamental information.

It mainly refers to the expectation of the expected resource service in the aspects of the basic information of the provider and the type of service provided.

(2) Expectation of service function.

It mainly includes the function expectation and post setting of service resources, the various resources that can be provided, the input parameters before the service starts, the output parameters after the completion of the service, and the result expectation after the completion of the task.

(3) Expectation of service performance.

It mainly includes three aspects: member evaluation index for each resource in the process of resource establishment, operation evaluation index in the operation process after resource establishment, and comprehensive evaluation index of resource. The performance of service resources is evaluated by setting the threshold of each evaluation index in the task ontology.

The second part of task ontology is task context information, which mainly refers to some special requirements of resource users for resource services due to their habitual

preferences or constraints of their own conditions during task execution. It mainly includes the following two aspects.

(1)    Fundamental information.

The fundamental information of task context information mainly refers to the limitation of service time and service cost intervals of resource users.

(2)    Dynamic information.

It refers to other constraints on tasks proposed by resource users due to their own conditions, such as their own site resources, geographical location, surrounding environment, local policies, personnel, etc. For example, due to the limitation of local environmental protection, only clean energy such as electric energy can be used to complete the casting in the designed production system.

Based on the preceding analysis of tasks, the task-ontology model is shown as follows.

$$\begin{cases} O_{Task} = \left( C_{Task}, \ P^C{}_{Task} \right) \\ C_{Task} = \left\{ Set_{SerExpe}, \ Set_{UserCntxt} \right\} \end{cases} \tag{1}$$

where, $O_{Task}$ represents the task-ontology model, represented by two tuples; $C_{Task}$ represents the set of task classes and describes the main information elements contained in tasks; $Set_{SerExpe}$ represents the set of classes that the task expects from the service, represented by the set shown in (2); $Set_{UserCntxt}$ represents the set of classes that the task describes to the user's own context, represented by the set shown in (3).

$$Set_{SerExpe} = \left\{ BscInf, \ Fcn, \ L_{Role}, \ L_{AuxRes}, \ L_{Input}, \ L_{Output}, \ L_{Res}, \ [Perf] \right\} \tag{2}$$

$$Set_{UserCntxt} = \left\{ TimRag, \ Cost, \ L_{DynCntxt} \right\} \tag{3}$$

In Formula (2), $BscInf$ represents the basic information of the expected service resource, mainly including the name, address, contact information and other information of the resource provider, as well as the type of service provided. $Fcn$ represents the expectation of service resource function. According to the size of task granularity, the function represented by $Fcn$ is quite different. $L_{Role}$ represents the list of design roles provided by the task for service resources, and it is the initiative resource needed to complete the task. $L_{Input}$ represents the list of task input parameters and is the initial parameter for the start of task. $L_{Output}$ represents the task output parameter list, which is the output list after the task is completed and represents the change of product task data information in the design process. $L_{Res}$ represents the output result list, which is the change of design status caused by design activities in the process of task execution. $[Perf]$ represents the evaluation index threshold array, which defines the threshold values of various resource evaluation indexes.

In Formula (3), $TimRag$ indicates the limit of service time required by the user. $Cost$ represents the limit on service price. $L_{DynCntxt}$ refers to the policy restrictions on dynamic environmental protection and energy in context information, as well as the user environment restrictions caused by personnel, technology and other factors.

## 4. Information Flow Based Design Task Decomposition

In the matching process of design task and cloud services, when the current cloud service or its composition cannot match the task directly, it is necessary to decompose the task to reduce the granularity of the task and improve the success rate of task matching. Therefore, the tasks published by resource requesters need to be decomposed so as to reduce the granularity of tasks and facilitate the search and matching of cloud services in the cloud manufacturing environment. Based on the study of information flow, a task-decomposition method based on information flow is proposed, which decomposes large-scale tasks and establishes the product development process with the fine granularity task.

### 4.1. Information Flow

Information flow is the reflection of the enterprise business process, which describes the information transmission and the characteristics of information during the task [52]. It is the integration of data flow and describes the input and output between tasks. The production and utilization of information in the process of product development often determines the quality and success of product development [53]. As information flow is easy to express the complex input and output information interactions, such as iteration and pre-release, the encapsulation and integration of task flow can be enhanced by using information flow in the design process.

In the process of product design, when a task is completed, the output information representing the change of data and the design result representing the change of state will be transferred to the node directed by the information flow. Information flow is an important tool for information exchange among resource providers in the product design process.

Five tuples are used to define the information flow in the task flow as follows

$$IfmFlw = (ID, \ Cont, \ Catgry, \ StrNod, \ EndNod) \tag{4}$$

where, $IfmFlw$ represents the information flow from $StrNod$ to $EndNod$; $Cont$ is the content of information flow, which reflects its purpose and intention of information flow; $Catgry$ indicates the type of information flow, which reflects its role in the task flow; $StrNod$ is the start node of information flow and $EndNod$ is the end node.

According to the different roles of information flow in the process of task execution, the information flow is divided into three categories, which are represented by sets as follows.

$$Catgry = \{Reg, \ Pre, \ Ite\} \tag{5}$$

where, $Reg$ represents regular information flow, which is used to represent the top-down information release between task nodes in the process of product development; $Pre$ stands for the pre-release information flow, and represents the advance release of information from the upstream task node to the downstream node; $Ite$ is the feedback information flow, and represents the feedback from downstream task node to upstream node, which will form the iteration of the node task.

With the introduction of the pre-release and feedback information flow, the product development process has the function of design iteration, which makes it possible to consider downstream design and manufacturing, analysis, manufacturability, assemblability and other quality issues in the process of upstream design. Increasing the number of iterations in a small range can reduce the large-scale design iterations, which can shorten the product development cycle and meet the needs of concurrent product development.

### 4.2. Node in Task Process

The design process is composed of a number of nodes. Among them, the node is either an indecomposable meta task or a subprocess composed of a group of meta tasks. Based on the preceding concepts, the task is encapsulated into a combination of several ordered nodes.

According to their different functions, the nodes are divided into three types: flag node, logical node and task node, which are represented as follows

$$Nodeset = \{Flg, \ Log, \ Tsk\} \tag{6}$$

where, $Flg$ is flag node and represents the start or end of the process; $Log$ indicates logical node, which expresses the composition relationship of tasks in the process; $Tsk$ is task node and indicates the meta task or the subprocess composed of meta tasks in the model.

According to different node types, the formal definition of each node is represented as follows

$$FlgNod = (ID, \; Catgry, \; IfwFlw, \; Cont) \tag{7}$$

$$LogNod = (ID, \; Catgry, \; Set_{Ifw}, \; Cont) \tag{8}$$

$$TskNod = (ID, \; Set_{InIfw}, \; Set_{OutIfw}, \; Set_{Task}, \; Set_{Role}, \; Set_{AuxRes}, TimeLim) \tag{9}$$

Formula (7) represents flag node, where $Catgry$ stands for the node type, which is divided into start node and end node, and its collection is defined as $FlgNodSet = \{Sta, \; End\}$. $IfwFlw$ represents information flow and stands for the information flow sent or received by this node. Formula (8) represents logical node, where $Catgry$ represents node type, and its value range is defined as shown in Formula (10). $Set_{Ifw}$ is the information flow set of a node and represents the information flow received and sent by this node. Its formal definition is shown in Formula (11), where $Ifw_i$ represents the No.$i$ information flow received or sent by this node.

$$Catgry = \{AndMrg, \; AndBif, \; OrMrg, \; OrBif\} \tag{10}$$

$$Set_{Ifw} = \{Ifw_1, \; Ifw_2, \dots, Ifw_n\} \tag{11}$$

Formula (9) represents task node, where $Set_{InIfw}$ and $Set_{OutIfw}$ represent the input and output information flow of the node respectively, which stands for the information input when the node starts and the information output when the node ends. $Set_{Task}$ represents the task set of this node, and stands for the corresponding meta task of the node, which is defined according to the needs of product design process. $Set_{Role}$ represents the role set required by this node and stands for the human resources required during the operation of the node. $Set_{AuxRes}$ represents the list of auxiliary resources required during the task execution of this node and $TimeLim$ indicates the time limit of this node task.

*4.3. Node Granularity and Task Process*

Node granularity is the number of indecomposable meta tasks contained in node tasks in the product design process. The larger the granularity of a node task, the more abstract the node is, and the more tasks it contains. Conversely, the more specific the node is, the less tasks it contains.

With the development of task and service resource matching, the granularity of a task is gradually reduced. From market research to conceptual design, to structural design and process design, the product development process model will become more and more detailed. Figure 5 shows the decomposition process of a product task. The granularity of the upper nodes in the diagram is large. With the decomposition of tasks, the model will be refined gradually, and the node granularity will be smaller.

(1)　Sequence process.

It is composed of a group of nodes without bifurcation, which is used to complete a series of serial product development processes, shown in Figure 6.

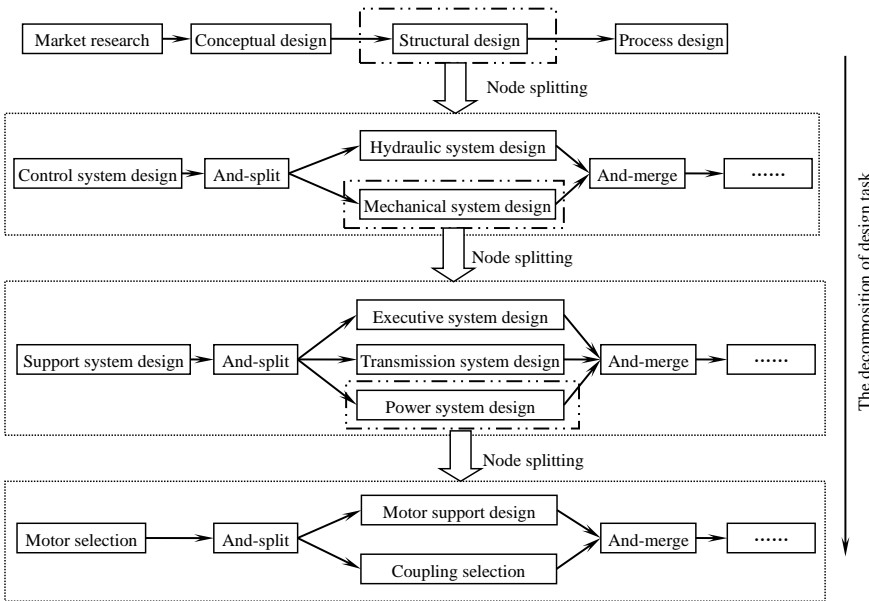

**Figure 5.** The decomposition process of product design process.

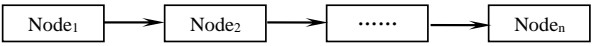

**Figure 6.** Sequence process.

(2)  Concurrent process.

Concurrent process is composed of multiple nodes without information coupling, and all the nodes proceed simultaneously. Only after all the tasks in the concurrent process are accomplished can the subsequent task start. The concurrent process starts with and-split node and ends with and-merge node, which is shown in Figure 7.

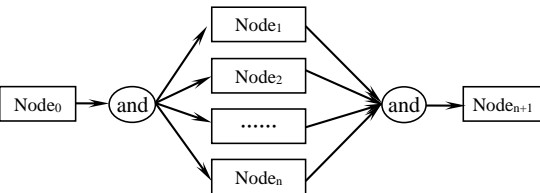

**Figure 7.** Concurrent process.

(3)  Selection process.

Selection process is composed of multiple nodes, one of which is performed when the task instance is executed. The path selection is determined by the attributes of tasks. The selection process starts with or-split node and ends with or-merge node, and its diagram is shown in Figure 8.

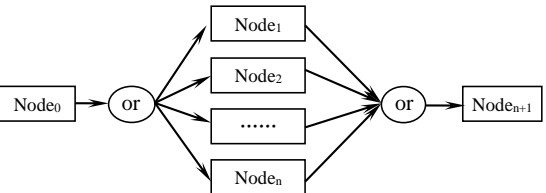

**Figure 8.** Selection process.

(4)  Iteration process.

Iteration process is a loop composed of one or multiple nodes, in which the activities are executed repeatedly until the given conditions are met. It is shown in Figure 9.

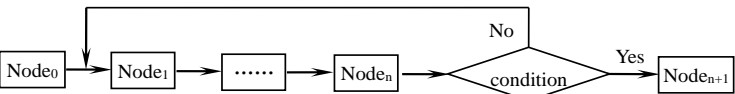

**Figure 9.** Iteration process.

*4.4. Information Flow-Based Task Decomposition*

For complex product design tasks involving multiple disciplines, there are few established CDUs that can directly match with them. Usually, it is necessary to decompose the interdisciplinary complex design tasks into meta-capability tasks in a single-discipline domain and then search for the design resources to match them. Therefore, with the development of the product design process, the nodes of the product development process will gradually split, and the tasks within the nodes will also be refined.

Node splitting means that with the advancement of the resource matching process, the coarse task nodes that fail to match are decomposed under the premise of retaining the original functions. Through node splitting, multiple fine granularity nodes and information flow among nodes are used to replace coarse granularity nodes to complete the matching of resources and tasks. The split nodes inherit the attributes of the upper node and realize its functions. Information flow is added between tasks of each split node to realize the input and output data interaction between nodes. The formal representation of task node splitting is as follows

$$TskNod = \sum_{i=1}^{m} TskNod_i + \sum_{j=1}^{n} LogNod_i + \sum_{k=1}^{l} FlgNod_k \tag{12}$$

where, *TskNod* represents the task node before splitting; $\sum_{i=1}^{m} TskNod_i$ is the *m* subtask node derived from splitting; $\sum_{j=1}^{n} LogNod_i$ stands for the *n* logical nodes derived from the process after splitting; $\sum_{k=1}^{l} FlgNod_k$ represents the split *l* flag nodes.

The definitions of the flag node and logical node generated in the above formula are shown as Formulas (7) and (8), and the derived task node is defined as follows

$$TskNod_i = (ID_i, Set_{InIfwi}, Set_{OutIfwi}, Set_{Taski}, Set_{Rolei}, Set_{AuxResi}, TimeLim_i) \tag{13}$$

where, *TskNod_i* stands for the No.*i* subtask node, and the elements in the formula represent its ID, input information flow set, output information flow set, task set, role set, auxiliary resource set and time limit respectively. Among them, the superposition of task set, role set, or auxiliary resource set is the corresponding element of its superior node, shown as follows

$$\begin{cases} Set_{Task} = \sum\limits_{i=1}^{m} Set_{Taski} \\ Set_{Role} = \sum\limits_{i=1}^{m} Set_{Rolei} \\ Set_{AuxRes} = \sum\limits_{i=1}^{m} Set_{AuxResi} \end{cases} \tag{14}$$



After task splitting, there are multiple relationships among subtask nodes, such as sequence, concurrent, iteration, etc. The relationship between the completion time limit of subtask nodes and the completion time limit of their superior nodes is shown as follows

$$
\begin{aligned}
TimeLim = & \sum_{i=1}^{s} TimeLim_i + \sum_{j=1}^{t} \left( TimeLim_{j1} \cap TimeLim_{j2} \cap \ldots \cap TimeLim_{jp} \right) \\
& + \sum_{k=1}^{r} \left( TimeLim_{k1} \cup TimeLim_{k2} \cup \ldots \cup TimeLim_{kq} \right)
\end{aligned}
\tag{15}
$$

where, $TimeLim$ represents the total time limit of the task; $TimeLim_i$ represents the time limit of the No. $i$ sequence node and s stands for the number of sequence node in the design process; $TimeLim_{jp}$ represents the time limit of the No. $p$ simultaneous branch of the No. $j$ concurrent process and $t$ stands for the number of concurrent process; $TimeLim_{kq}$ represents the time limit of the No. $q$ iteration of the No. $k$ iteration process and $r$ stands for the number of iteration process.

The set of information flow in Formula (13) includes not only the elements inherited from the superior node but also the information flow transmitted within each node after splitting. Its formal definition is as follows

$$
\begin{cases}
Set_{InIfwi} = Set_{InIfwk} + \sum_{j=1}^{n} Set_{OutIfwj} \\
Set_{InIfw} = \sum_{k=1}^{n} Set_{InIfwk}
\end{cases}
\tag{16}
$$

where, $Set_{InIfwi}$ represents the set of all the input information flow; $Set_{InIfwk}$ represents the input information flow from its parent node after node splitting; $Set_{OutIfwj}$ represents the set of output information flows of each node after splitting. The flow direction is determined according to the attributes of the output information flow. If it flows to other nodes after splitting, it will be recorded. If it flows to other nodes than the node generated by splitting, it will be discarded. When $i = j$, the output information flow is the feedback information flowing to itself.

Similarly, the formal definitions of $Set_{OutIfw}$ and $Set_{OutIfwi}$ are as follows

$$
\begin{cases}
Set_{OutIfwi} = Set_{OutIfwk} + \sum_{j=1}^{n} Set_{OutIfwj} \\
Set_{OutIfw} = \sum_{k=1}^{n} Set_{OutIfwk}
\end{cases}
\tag{17}
$$

where, $Set_{OutIfwi}$ represents all the input information flow after node splitting; $Set_{OutIfwk}$ represents the output information flow from the child node to the parent node after node splitting; $Set_{OutIfwj}$ represents the information flow passed to its parent node after splitting. When $i = j$, the output information flow is the feedback information flowing to itself.

According to the preceding discussion on the information flow, the transmission of information flow in the process of node splitting is shown in Figure 10.

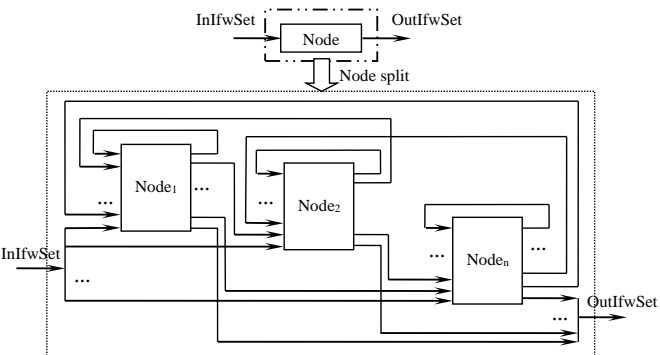

**Figure 10.** The information flow in node splitting.

## 5. Ontology Task and Cloud Service Matching Strategy Based on Semantic Similarity

Based on the basic algorithm of semantic similarity, a multistage matching algorithm between ontology tasks and cloud service is proposed, which is used to complete the matching between tasks and cloud services.

### 5.1. Semantic Similarity Basic Algorithm

The main content of semantic similarity calculation includes text and numerical values, and their basic algorithm is as follows.

(1)    Text-phrase similarity matching algorithm.

With the help of a natural language-processing method, the semantic matching of service resources is carried out by comparing the similarity of text description information in ontology concept elements in terms of word form or word meaning, which is the most direct method of ontology matching. Zhang [54] proposes a text-similarity algorithm. Firstly, remove the stop words in the ontology concept, count the frequency of each word in the text and extract the keywords, and then calculate the similarity of the keyword groups.

Suppose there are two key phrases, $Res_1$ and $Res_2$, which are composed of $m$ and $n$ keywords, respectively. The similarity calculation process is as follows.

Step 1: With $Res_1$ as the outer loop and $Res_2$ as the inner loop, a double nested loop structure is established.

Step 2: For each keyword in $Res_1$, the maximum word similarity with $Res_2$ is calculated in the inner loop, which is $word1Sim_i$ ($i = 1, 2, \ldots, m$).

Step 3: Add the maximum word similarity calculated in step 2, and the result is $word1sum = \sum_{i=1}^{m} word1sum_i$.

Step 4: Get the variable: $word1Sim = word1Sim/m$.

Step 5: Similarly, $Res_2$ is used as the outer loop and $Res_1$ as the inner loop. For each keyword in $Res_2$, the maximum word similarity with $Res_1$ is calculated as follows: $word2Sim_j$ ($j = 1, 2, \ldots, n$)

Step 6: Add the maximum word similarity calculated in step 5 to achieve the result $word2sum = \sum_{j=1}^{n} word2sum_j$.

Step 7: Get the variable: $word2Sim = word2Sim/n$.

Step 8: Taking the mean value of $word1Sim$ and $word2Sim$, the similarity of $Res_1$ and $Res_2$ is as follows: $wordSim = (word1Sim + word2sim)/2$.

The pseudo code for phrase similarity calculation is shown as Algorithm 2.

**Algorithm 2.** The Algorithm of Phrase Similarity Calculation Process

---

1. **Input:** key phrases $Res_1$ and $Res_2$
    $Res_1 = \sum_{i=1}^{m} word1_i$
    $Res_2 = \sum_{j=1}^{n} word2_j$
2. **Output:** $wordsim(Res_1, Res_2)$, the similarity of phrases $Res_1$ and $Res_2$
3. **for** $(i = 0; i < m; i\,{+}{+})$ **do**
4. **for** $(j = 0; j < n; j\,{+}{+})$ **do**
5. $word1sim_i \leftarrow$ the maximum similarity between $word1_i$ and $word2_j$
6. **end for**
7. $word1sim = word1sim + word1sim_i$
8. **end for**
9. $word1sim = word1sim/m$
10. **for** $(i = 0; i < m; i\,{+}{+})$ **do**
11. **for** $(j = 0; j < n; j\,{+}{+})$ **do**
12. $word2sim_i \leftarrow$ the maximum similarity between $word1_i$ and $word2_j$
13. **end for**
14. $word2sim = word2sim + word2sim_i$
15. **end for**
16. $word2sim = word2sim/m$
17. $wordsim(Res_1, Res_2) = (word1sim + word2sim)/2$
18. **return**

---

The fundamental idea of the preceding algorithm is to average the maximum similarity of the two keyword phrases, which traverses all the keywords in the keyword group and is not sensitive to the order of keywords. In this algorithm, the traversal times of a single nested loop is $m \times n$, and the total time complexity is $2(m \times n)$.

(2)   Sentence-similarity matching algorithm.

The preceding text-phrase similarity matching algorithm can only calculate the similarity of keywords, which is completed by extracting keywords from the text and comparing them. It cannot judge the length of the text and the order of keywords in the text. For the ontology expressed in the form of sentences, the algorithm in [33] is utilized to judge the similarity of two sentences in terms of word shape, sentence length and order, and then it is synthesized into sentence similarity according to a certain weight. When the similarity value reaches or exceeds the preset threshold, the two sentences are considered to be similar.

Suppose that $sent_1$ and $sent_2$ are sentences describing the concepts of ontology. $SET_i$ and $SET_j$ are used to represent the ordered set of all the words, and the union of the above two sets is $SET_{or} = \{U_1, U_2, \ldots, U_m\}$. The morphological similarity, order similarity and length similarity are calculated as follows.

Step 1: Sentence morphological similarity calculation.

$U_k(0 < k \leq m)$ is any word in $SET_{or}$. Suppose that the maximum similarity between $U_k$ and all words in $SET_i$ is $SIM_{ik}$, and the maximum similarity with all words in $SET_j$ is $SIM_{jk}$. The morphological similarity between $sent_1$ and $sent_2$ is shown as follows

$$SIM_1(sent1, sent2) = 1 - \frac{\left|\sum_{k=1}^{m} SIM_{ik} - \sum_{k=1}^{m} SIM_{jk}\right|}{\sum_{k=1}^{m} SIM_{ik} + \sum_{k=1}^{m} SIM_{jk}} \tag{18}$$

Step 2: Sentence order similarity calculation.

Suppose that $Ser_i$ represents the position sequence number of the word in $SET_i$, and $Ser_j$ represents the position sequence number of the word in $SET_j$. Then the order similarity between $sent_1$ and $sent_2$ is shown as follows

$$SIM_2(sent1, sent2) = 1 - \frac{\left||Ser_i| - |Ser_j|\right|}{|Ser_i| + |Ser_j|} \tag{19}$$

Step 3: Sentence length similarity calculation.

If the lengths of sentences $sentence_1$ and $sentence_2$ are $len_i$ and $len_j$ respectively, the length similarity between the two sentences can be shown as follows

$$SIM_3(sent1, sent2) = 1 - \frac{\left|len_i - len_j\right|}{len_i + len_j} \tag{20}$$

Step 4: Sentence similarity calculation

Set the weight of word shape similarity, order similarity and sentence length similarity in sentence similarity judgment as $\omega_1$, $\omega_2$ and $\omega_3$, respectively, then the calculation formula of similarity between $sent_1$ and $sent_2$ is as follows

$$
\begin{aligned}
&SenSim(sent1, sent2) \\
&= \omega_1 \times \left(1 - \frac{\left|\sum_{k=1}^{m} SIM_{ik} - \sum_{k=1}^{m} SIM_{jk}\right|}{\sum_{k=1}^{m} SIM_{ik} + \sum_{k=1}^{m} SIM_{jk}}\right) + \omega_2 \times \left(1 - \frac{\left|\left|Ser_i\right| - \left|Ser_j\right|\right|}{\left|Ser_i\right| + \left|Ser_j\right|}\right) + \omega_3 \times \left(1 - \frac{\left|len_i - len_j\right|}{len_i + len_j}\right)
\end{aligned} \tag{21}
$$

(3) Numerical-interval similarity matching algorithm.

Li [24] proposed a method for calculating the similarity of numerical interval. Suppose that $Para_i$ is a numerical parameter of the task, $Para_j$ is the corresponding numerical parameter of the service resource to be selected, and $Lgh_i$ and $Lgh_j$ are the numerical intervals of the two numerical parameters, then the similarity between the two parameters is expressed as follows

$$
ValSim(Para_i, Para_j) = \begin{cases} 0 & (Lgh_i \cap Lgh_j = \Phi) \\ \frac{\left|Lgh_i \cap Lgh_j\right|}{\left|Lgh_i\right|} & (Lgh_i \cap Lgh_j \neq \Phi \text{ and } Lgh_i \cap Lgh_j \neq Lgh_i) \\ 1 & (Lgh_i \cap Lgh_j = Lgh_i \text{ or } Lgh_i \cap Lgh_j = Lgh_j) \end{cases} \tag{22}
$$

*5.2. Semantic Similarity Based Multistage Matching Strategy between Ontology Task and Cloud Service*

The multistage resource-modeling method proposed in [3] is used to accomplish the design resource modeling, and then the resource ontology is transformed into cloud service by using the semantic-based resource-servitization method. Through the mapping between resource ontology and OWL-S extended ontology, resource servitization is accomplished and cloud services based on semantics are created [48]. For ontology-based task and cloud service matching research, its essence is to calculate the semantic similarity between published services and tasks. The higher the similarity between tasks and services, the higher the matching degree between them.

Task ontology and resource service ontology are expressed by concepts, so service matching becomes the similarity comparison of elements in an ontology concept set. Although the composition of concepts is complex, it can be divided into two categories: text and numerical value. The former mainly refers to phrases and sentence descriptions in the form of text, and the latter refers to the measurement expressed by quantitative value or numerical interval, or fuzzy number that can be converted into quantitative value.

Based on the matching of basic elements such as text, sentence and value, which constitute the ontology concept, and taking the importance of ontology elements in the process of service resource matching as the sequence, the multistage matching strategy of task ontology and service ontology is constructed, shown in Figure 11.

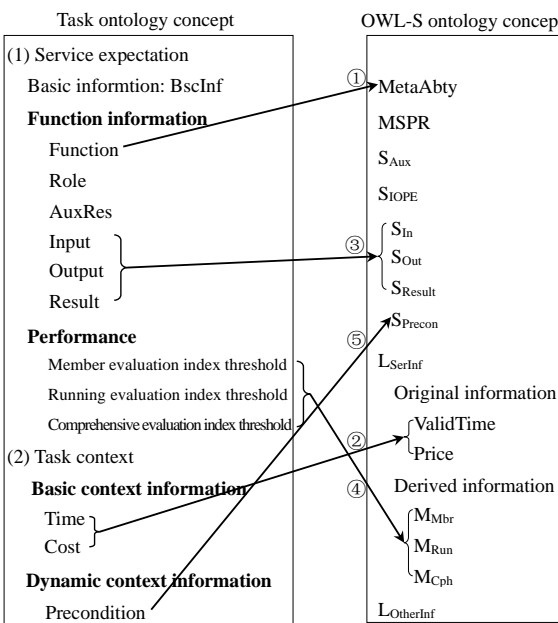

**Figure 11.** The multistage matching between task and service.

According to the importance of ontology elements, five levels of matching, including task and service resource function matching, task context matching, input, output and result (IOR) matching, evaluation index matching and precondition matching, are carried out in turn, which is shown in Figure 12. The steps of multistage service resource matching based on semantic similarity are as follows.

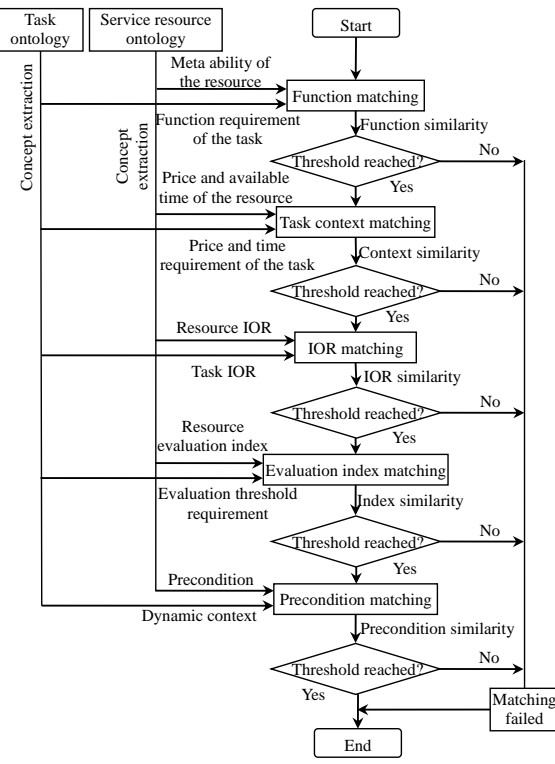

**Figure 12.** Multistage matching flow chart between task and service.

(1)    Function matching.

This is the matching of function description in the request task ontology and resource service ontology. The function description is usually represented by unordered phrases.

If the phrase set describing the function of task ontology is *Task.Mission* and the function of resource service ontology is described as *Serv.Function*, then the matching similarity is shown as follows

$$Match_{Fcn}(Task.Mission, \ Serv.Function) = WordSim(Task.Mission, \ Serv.Function) \quad (23)$$

(2)   Task context matching

Task context consists of time and cost. Through operation, the matching of these two parts can be transformed into the matching of numerical intervals. Assuming that the required time and acceptable cost regions of the task are *Task.Time* and *Task.Cost*, respectively, and the time and cost regions that resource services can provide are *Serv.Time* and *Serv.Pric*, the task context matching degree is as follows

$$Match_{Cnt}(Task.Context, \ Serv.SerInf) = \alpha \times ValSim(Task.Time, \ Serv.Time) + \beta \times ValSim(Task.Cost, \ Serv.Pric) \quad (24)$$

(3)   IOR matching

The parameter of IOR, consisting of input parameter sequence, output parameter sequence and result sequence, are reordered according to their composition form, which are combined into *m* text phrase set *WSetIor*, *n* sentence set *SSetIor* and *l* numerical interval set *VSetIor*. The IOR matching degree is as follows

$$
\begin{aligned}
Match_{IOR}&(Task.IOR, \ Serv.IOR) \\
&= \sum_{i=1}^{m} (\omega_i \times WordSim(Task.WSetIor_i, \ Serv.WSetIor_i)) \\
&+ \sum_{j=1}^{n} (\omega_j \times SenSim(Task.SSetIor_j, \ Serv.SSetIor_j)) \\
&+ \sum_{k=1}^{l} (\omega_k \times ValSim(Task.VSerIor_k, \ Serv.VSetIor_k))
\end{aligned} \quad (25)
$$

where, $\omega_i$ is the weight coefficient of the No. *i* matching phrase; $\omega_j$ is the weight coefficient of the No. *j* matching sentence; $\omega_k$ is the weight coefficient of the No. *l* numerical interval.

(4)   Evaluation index matching.

Evaluation index refers to the index generated by the evaluation variables of resources during the establishment or operation of service resources, which is directly read from the evaluation matrix [3]. By comparing the resource evaluation index with the threshold set in the task ontology, it can be judged whether the service resources meet the requirements. The weighted average of design capability index, customer evaluation index and cost index of service resources is taken as the similarity of evaluation index, as shown below

$$Match_{Eva}(Task.Eva, \ Serv.Eva) = \alpha \times M_{Cph}.Abi + \beta \times M_{CPH}.Eva + \gamma \times M_{Cph}.Cost \quad (26)$$

where, $M_{Cph}.Abi$, $M_{CPH}.Eva$ and $M_{Cph}.Cost$ is the design capability index, customer evaluation index and cost index of service resources; $\alpha$, $\beta$ and $\gamma$ are their weights.

(5)   Precondition matching.

Similar to IOR matching in (3), preconditions can be divided into several sets of phrases, sentences and numerical intervals, which are respectively expressed as *m* phrase

set *WSetPre*, *n* sentence set *SSetPre* and *l* numerical interval set *VSerPre*. The matching degree of preconditions is shown as follows

$$
\begin{aligned}
Match_{Pre}(&Task.Pre,\ Serv.Pre) \\
&= \sum_{i=1}^{m} (\omega_i \times WordSim(Task.WSetPre_i,\ Serv.WSetPre_i)) \\
&+ \sum_{j=1}^{n} (\omega_j \times SenSim(Task.SSetPre_j,\ Serv.SSetPre_j)) \\
&+ \sum_{k=1}^{l} (\omega_k \times ValSim(Task.VSerPre_k,\ Serv.VSetPre_k))
\end{aligned}
\tag{27}
$$

where, $\omega_i$, $\omega_j$ and $\omega_k$ are the weight coefficient of matching phrase, sentence and numerical interval respectively.

The pseudo code of preceding multistage matching between ontology tasks and service resources based on semantic similarity is shown as Algorithm 3.

---

**Algorithm 3.** The Algorithm of Multistage Matching between Ontology Task and Cloud Service

---

1. **Input:** design task ontology *TO*
        cloud service ontology *SO*
2. **Output:** the matching result between *TO* and *SO*
3. *TO.Mission* ← task requirement abstraction from *TO*
4. *SO.Function* ← service function abstraction from *SO*
5. Matching between *TO.Mission* and *SO.Function*
6. **if** (function similarity threshold reached) **then**
7. *TO.Context* ← task context abstraction from *TO*
8. *SO.SerInf* ← service time & cost information abstraction from *SO*
9. Matching between *TO.Context* and *SO.SerInf*
10. **if** (context similarity threshold reached) **then**
11. *TO.IOR* ← task I&O abstraction from *TO*
12. *SO.IOR* ← service I&O abstraction from *SO*
13. Matching between *TO.IOR* and *SO.IOR*
14. **if** (IOR similarity threshold reached) **then**
15. *TO.ThreRequ* ← task evaluation index requirement abstraction from *TO*
16. *SO.EvalIndex* ← service evaluation index abstraction from *SO*
17. Matching between *TO.ThreRequ* and *SO.EvalIndex*
18. **if** (evaluation index similarity threshold reached) **then**
19. *TO.DynaContext*← task dynamic context abstraction from *TO*
20. *SO.Promise* ← service precondition abstraction from *SO*
21. Matching between *TO.DynaContext* and *SO.Promise*
22. **if** (promise similarity threshold reached) **then**
23. matching succeed between *TO* and *SO*
24. **end if**
25. **else**
26. matching failed
27. **end if**
28. **else**
29. matching failed
30. **end if**
31. **else**
32. matching failed
33. **end if**
34. **else**
35. matching failed
36. **end if**
37. **return**

---

## 6. Intelligent Optimization of Cloud Services

When there are enough candidate cloud services, there will be multi-cloud services matching with ontology tasks, which meet the expected threshold requirements. In order to accomplish the design of complex products efficiently, the improved differential evolution algorithm is used to further optimize the cloud services that meet the threshold requirements, so as to obtain the optimal service composition.

The basic differential evolution algorithm mainly includes the steps of initialization, mutation, crossover and selection [55]. In this algorithm, a random initial assignment is made to the population, which is usually $N_p$ d-dimensional vectors. The initialization value of population is generally selected randomly from the values within the given boundary constraint. In terms of calculation process, differential evolution algorithm is similar to the real coded genetic algorithm, including the operation of crossover, mutation and selection. However, in the way of generation, the two algorithms are different. Differential evolution algorithm generates mutation individuals based on the difference vector between parents, and then cross operates the parent and mutation individuals according to a certain probability, and finally uses greedy selection strategy to generate offspring individuals [56].

An improved differential evolution algorithm [45] is used to optimize the composition of cloud services, which is based on the basic differential evolution algorithm [57], and its evolution performance has been improved and meets the actual production requirements.

### 6.1. Parameter and Fitness Function of the Optimization

In the process of optimizing the composition of tasks and cloud services, the parameters are set as follows.

(1)　Task parameters.

Task parameters include total task *Tol*, subtask module set and total subtask. Design subtask model set *ST* is shown as follows

$$ST = \{st_1, st_2, \ldots, st_i, \ldots, st_N\} \tag{28}$$

where, $st_i$ is subtask module, $i = 1, 2, \ldots, N$. $N$ is the total number of subtask modules after decomposition.

Assuming that all tasks include each subtask module, the task amount $Tol_i$ of the subtask module is shown as follows

$$\sum_{i=1}^{N} Tol_i = Tol \tag{29}$$

where, $Tol_i$ is the number of the No.*i* subtask module, $i = 1, 2, \ldots, N$.

(2)　Candidate cloud service set of subtask module.

The candidate cloud service set of subtask module $st_i$ is shown as follows

$$CS^i = \left\{cs_1^i, \ cs_2^i, \ldots, cs_j^i, \ldots, cs_{M_i}^i\right\} \tag{30}$$

where, $cs_j^i$ is the No.*j* candidate cloud service of subtask module $st_i$; $j = 1, 2, \ldots, M_i$; $M_i$ is the total number of candidate cloud services.

(3)　Candidate cloud service time consumption.

It includes two parts, and the first is the time consumption for cloud service $cs_j^i$ to complete subtask module $st_i$, shown as follows

$$T_j^i = t_j^i \times x_j^i \tag{31}$$

where, $t_j^i$ is the time consumption that candidate cloud service $cs_j^i$ accomplishes unit task module $st_i$; $x_j^i$ is the number of tasks assigned by subtask module $st_i$ to candidate cloud service $cs_j^i$, and $\sum_{j=1}^{j=M_i} x_j^i = Tol_i$.

The second part of the candidate cloud service time consumption is the handover time consumption $t_{j \to j'}^i$ between two adjacent cloud services, $cs_j^i$ of subtask module $st_i$ and $cs_{j'}^{i+1} (j = 1, 2, \ldots, M_i, j' = 1, 2, \ldots, M_{i+1})$ of subtask module $st_{i+1}$.

Taking the shortest product design time consumption as the goal of cloud service optimization, the fitness function of optimization is as follows

$$\min_{min} F = T_{max} \tag{32}$$

where, $T_{max}$ is the longest time consumption required for tasks to be delivered to the service requester.

### 6.2. Algorithm Design

#### 6.2.1. Improved Differential Evolution Algorithm

In order to adapt to the actual working condition of optimal service composition in a cloud manufacturing system, we utilize the method of gene position division based on a subtask module in coding mode [47], which decomposes the whole task into different modules, and all the operation, such as mutation, crossover and selection, are accomplished within the module.

(1)    Coding method.

The bijective relationship between each gene coding and the corresponding candidate cloud services is established by using the real number coding in the range of $[0, Tol_i]$. The candidate cloud service resources are coded according to the subtask module, and the number of gene bits of each subtask module is the number of candidate cloud service. Therefore, the gene bits of candidate cloud service $cs_m^n$ is as follows

$$cs_m^n = \begin{cases} m + \sum\limits_{i=1}^{n-1} M_i & n > 1 \\ m & n = 1 \end{cases} \tag{33}$$

where, $cs_m^n$ represents the No. $m$ candidate cloud service of subtask module $st_n$, $n$ stands for the No. $n$ subtask module, and $m$ represents the No. $m$ candidate cloud service of this subtask module.

(2)    Block mutation operation.

In this improved differential evolution method, the mutation operation is partitioned according to the subtask module. Suppose that the No.$h$ generation population of the No.$n$ subtask module is $X^{h,n} = \left\{ x_d^{h,n} \mid d = 1, 2, \ldots, N_p \right\}$, the gene sequence of the No. $d$ chromosome of this population is $x_d^{h,n} = \left\{ x_{d,1}^{h,n}, x_{d,\,2}^{h,n}, \ldots, x_{d,M_n}^{h,n} \right\}$. Where, $N_p$ is the population size.

Block mutation is performed on the No.$d$ chromosome. Three natural numbers $d_1$, $d_2$ and $d_3$, which are not equal to $d$ and are in the range of $\left[1, \ N_p\right]$, are generated randomly. The new mutants are $u_d^{h,n} = \left\{ u_{d,1}^{h,n}, u_{d,\,2}^{h,n}, \ldots, u_{d,M_n}^{h,n} \right\}$. The calculation formula of mutation individual is as follows

$$u_{d,\,i}^{h,n} = x_{d_1,\,i}^{h,n} + \mathrm{F} \cdot \left( x_{d_2,\,i}^{h,n} - x_{d_3,\,i}^{h,n} \right) \tag{34}$$

where, F is a real constant and represents the scaling factor of the No.$n$ module. Its value is 0.5.

(3)    Block crossover operation.

Within the subtask module, the cross operation between the parent individual $x_d^{h,n}$ and the mutant individual $u_d^{h,n}$ is carried out, which results in experimental individual $v_d^{h,n} = \{v_{d,1}^{h,n}, v_{d,2}^{h,n}, \dots, v_{d,M_n}^{h,n}\}$. The crossover operation is as follows

$$v_{d,i}^{h,n} = \begin{cases} u_{d,i}^{h,n} & r_i \leq CR \cap i = \text{rf} \\ x_{d,i}^{h,n} & \text{others} \end{cases} \tag{35}$$

where, $r_i$ is the No.$i$ random real number between [0, 1], and rf is a natural number between [1, $Mn$], which ensure that $v_d^{h,n}$ achieves at least one gene value from $u_d^{h,n}$. CR is the crossover operator, and its value is 0.5.

It is necessary to check and correct $v_{d,i}^{h,n}$ to ensure that it can meet the requirement: $\sum_{i=1}^{M_n} v_{d,i}^{h,n} = Tol_n$. If $\sum_{i=1}^{M_n} v_{d,i}^{h,n} > Tol_n$, reduce the task number of candidate cloud service, which can reduce the time consumption of the subtask. If $\sum_{i=1}^{M_n} x_j^i < Tol_n$, increase the task number of candidate cloud service, which can increase the time consumption. The process is as follows.

Step 1: Round the experimental individual $v_{d,i}^{h,n}$ and calculate the time consumption for each cloud service node based on formula $T_{d,i}^{h,n} = t_{d,i}^{h,n} \times v_{d,i}^{h,n}$. If $\sum_{i=1}^{M_n} v_{d,i}^{h,n} > Tol_n$, turn to Step 2; if $\sum_{i=1}^{M_n} v_{d,i}^{h,n} < Tol_n$, turn to Step 4; if $\sum_{i=1}^{M_n} v_{d,i}^{h,n} = Tol_n$, end the operation.

Step 2: According to $T_{d,i}^{h,n}$, the most time-consuming cloud service node, $I_{max}$, is obtained. The task amount of $I_{max}$ is reduced by 1, that is, $v_{d,I_{max}}^{h,n} = v_{d,I_{max}}^{h,n} - 1$.

Step 3: Calculate the value of $\sum_{i=1}^{M_n} v_{d,i}^{h,n}$ exceeding $Tol_n$, which is named F. If F $\neq$ 0, turn to Step 2; if F = 0, end the operation.

Step 4: According to $T_{d,i}^{h,n}$, the least time-consuming cloud service node, $I_{min}$, is obtained. The task amount of $I_{min}$ is increased by 1, that is, $v_{d,I_{max}}^{h,n} = v_{d,I_{max}}^{h,n} + 1$.

Step 5: Calculate the value of $Tol_n$ exceeding $\sum_{i=1}^{M_n} v_{d,i}^{h,n}$, which is named $F$. If $F \neq 0$, turn to Step 4; if $F = 0$, end the operation.

The pseudo code for block crossover operation is shown as Algorithm 4.

(4)    Block selection operation.

A selection operation, using greedy algorithm, is performed between the parent individual $x_d^h$ and the experimental individual $v_d^h$, by which the offspring individual is generated. The selection formula of offspring individual $x_d^{h+1}$ is as follows

$$x_d^{h+1} = \begin{cases} v_d^h & T_{max}^v < T_{max}^x \\ x_d^h & \text{others} \end{cases} \tag{36}$$

where, $v_d^h$ is the experimental individual, $x_d^h$ is the parent individual, and $x_d^{h+1}$ is offspring individual. $T_{max}^v$ and $T_{max}^x$ are the maximum time consumption required for each batch of subtasks to be delivered to the service requesters.

---

**Algorithm 4.** The Algorithm of Block Crossover Operation

---

1. **Input:** experimental individual $v_d^{h,n} = \left\{ v_{d,1}^{h,n}, v_{d,2}^{h,n}, \ldots, v_{d,M_n}^{h,n} \right\}$

2. **Output:** changed experimental individual $v_{d,i}^{h,n}$ $i = 1, 2, \ldots, M_n$

3. **if** $(\sum_{i=1}^{M_n} round\left( v_{d,i}^{h,n} \right) > \mathrm{Tol}_n)$ **then**

4. continue

5. **else if** $(\sum_{i=1}^{M_n} round\left( v_{d,i}^{h,n} \right) < \mathrm{Tol}_n)$ **then go to** 21

6. **else return**

7. **end if**

8. $T_{Max} \leftarrow 0, x \leftarrow 0$

9. **for** $(i = 0; i < M_n; i$ ++$)$ **do**

10. $T_i \leftarrow t_{d,i}^{h,n} \times v_{d,i}^{h,n}$

11. **if** $(T_{Max} < T_i)$ **then**

12. $T_{Max} \leftarrow T_i$

13. $x \leftarrow i$

14. **end if**

15. **end for**

16. $v_{d,x}^{h,n} = v_{d,x}^{h,n} - 1$

17. $F \leftarrow \sum_{i=1}^{M_n} v_{d,i}^{h,n} - \mathrm{Tol}_n$

18. **if** $(F == 0)$ **then**

19. **return**

20. **end if**

21. $T_{Min} \leftarrow 0, y \leftarrow 0$

22. **for** $(i = 0; i < M_n; i$ ++$)$ **do**

23. $T_i \leftarrow t_{d,i}^{h,n} \times v_{d,i}^{h,n}$

24. **if** $(T_{Min} > T_i)$ **then**

25. $T_{Min} \leftarrow T_i$

26. $y \leftarrow i$

27. **end if**

28. **end for**

29. $v_{d,y}^{h,n} = v_{d,y}^{h,n} + 1$

30. $F \leftarrow \sum_{i=1}^{M_n} v_{d,i}^{h,n} - \mathrm{Tol}_n$

31. **if** $(F <> 0)$ **then**

32. **go to** 21

33. **end if**

34. **return**

---

### 6.2.2. Handover Strategy Design

According to the preceding improved differential evolution algorithm, there are a variety of cloud service composition schemes, and each of them has multiple handover schemes. In other words, the same resource can be handed over to multiple candidate cloud services of the next subtask module in various ways. Therefore, it is necessary to find the least time-consuming handover scheme in the same cloud service composition scheme, so that the product design time consumption of cloud service composition scheme is the shortest. Suppose that the handover scheme from the No. $n$ subtask module to the No. $n + 1$ subtask module is $TRANS_{M_n \times M_{n+1}}$. The handover process is as follows.

Step 1: The task allocation scheme of candidate cloud services, which is used to accomplish subtask module $st_n$ and $st_{n+1}$, are obtained from the genome, and we obtain two arrays $X^n = \{x_1^n, x_2^n, \ldots, x_{M_n}^n\}$ and $X^{n+1} = \{x_1^{n+1}, x_2^{n+1}, \ldots, x_{M_{n+1}}^{n+1}\}$. The tasks number of both arrays are *Tol*, that is, $Tol_n = Tol_{n+1} = Tol$. All the time consumption, which are used to hand over the task from candidate cloud service $cs_j^n$ to $cs_k^{n+1}$, are obtained from cloud manufacturing platform to establish the handover time consumption matrix $T_{M_n \times M_{n+1}}^{n \to n+1}$.

Step 2: Initialize the handover matrix $TRANS_{M_n \times M_{n+1}}$, and assign each element to 0, that is $trans_{i \to j} = 0$, where $i = 1, 2, \ldots, M_n$ and j $= 1, 2, \ldots, M_{n-1}$. In this initial state, the handover number of each path from subtask module $st_n$ to next module $st_{n+1}$ is 0.

Step 3: The elements in $T^{n \to n+1}_{M_n \times M_{n+1}}$ are arranged in ascending order, and the starting and ending cloud services after arrangement are recorded with set $H$. $H = \{h_k = (s_k, e_k) \mid s_k = 1, 2, \ldots, M_n; e_k = 1, 2, \ldots, M_{n+1}; k = 1, 2, \ldots, M_n \times M_{n+1}\}$, $|H| = M_n \times M_{n+1}$. Let $k = 1$.

Step 4: If $M_n \times M_{n+1} > k$, continue to perform the following operations, otherwise end the process. If $x^n_{s_k} \neq 0$ and $x^{n+1}_{e_k} \neq 0$, select the smaller of $x^n_{s_k}$ and $x^{n+1}_{e_k}$. If $x^n_{s_k} < x^{n+1}_{e_k}$, then $x^{n+1}_{e_k} = x^{n+1}_{e_k} - x^n_{s_k}$, $Tol_n = Tol_n - x^n_{s_k}$, $Tol_{n+1} = Tol_{n+1} - x^n_{s_k}$. If $x^n_{s_k} \geq x^{n+1}_{e_k}$, then $x^n_{s_k} = x^n_{s_k} - x^{n+1}_{e_k}$, $Tol_n = Tol_n - x^{n+1}_{e_k}$, $Tol_{n+1} = Tol_{n+1} - x^{n+1}_{e_k}$.

Step 5: If $Tol_n = 0$ or $Tol_{n+1} = 0$, then end the process; otherwise $k = k + 1$, and turn to Step 4.

The pseudo code for handover process is shown as Algorithm 5.

---

**Algorithm 5.** The Algorithm of Handover Process

---

1. **Input:** handover scheme from No.$n$ subtask module to No. $n + 1$ subtask module $TRANS_{M_n \times M_{n+1}}$
2. **Output:** least time-consuming handover scheme
3. $X^n \leftarrow \{x^n_1, x^n_2, \ldots, x^n_{M_n}\}$
4. $X^{n+1} \leftarrow \{x^{n+1}_1, x^{n+1}_2, \ldots, x^{n+1}_{M_{n+1}}\}$
5. $T^{n \to n+1}_{M_n \times M_{n+1}} \leftarrow$ hand over time consumption from $cs^n_j$ to $cs^{n+1}_k$
6. **for** ($i = 0$; $i < M_n$; $i$ ++) **do**
7.     **for** ($j = 0$; j $< M_{n-1}$; $j$ ++) **do**
8.     $trans_{i \to j} \leftarrow 0$
9.       **end for**
10. **end for**
11. arrange $T^{n \to n+1}_{M_n \times M_{n+1}}$ in ascending order
12. **while** ($Tol_n <> 0$ *or* $Tol_{n+1} <> 0$)
13. $k \leftarrow k+1$
14. **if** ($M_n \times M_{n+1} > k$) **then**
15. **if** ($x^n_{s_k} <> 0$ and $x^{n+1}_{e_k} <> 0$) **then**
16.    $trans_{s_k, e_k} \leftarrow \min(x^n_{s_k}, x^{n+1}_{e_k})$
17. **end if**
18. **if** ($x^n_{s_k} < x^{n+1}_{e_k}$) **then**
19.    $x^{n+1}_{e_k} \leftarrow x^{n+1}_{e_k} - x^n_{s_k}$
20.    $Tol_n \leftarrow Tol_n - x^n_{s_k}$
21.    $Tol_{n+1} \leftarrow Tol_{n+1} - x^n_{s_k}$
22.    $x^n_{s_k} \leftarrow 0$
23. **else**
24.    $x^n_{s_k} \leftarrow x^n_{s_k} - x^{n+1}_{e_k}$,
25.    $Tol_n \leftarrow Tol_n - x^{n+1}_{e_k}$
26.    $Tol_{n+1} \leftarrow Tol_{n+1} - x^{n+1}_{e_k}$
27.    $x^{n+1}_{e_k} \leftarrow 0$
28. **end if**
29. **end while**
30. **return**

---

### 6.2.3. Calculation of Fitness Function

When the cloud service composition is executed, its running rules are as follows.

(1) Each cloud service composition scheme can be regarded as being composed of several concurrent and intersecting design paths, and the time consumption of the most time-consuming design path represents the design time consumption of the scheme.

(2) In order to reduce the complexity of handover, each candidate cloud service of the subtask is handed over to the next candidate cloud service after completing its own task.

Based on the above rules, the fitness function is calculated as follows.

Step 1: The time consumption matrix $T_{\text{end}}$ of each candidate cloud service is established and initialized to 0.

Step 2: If $n = 1$, calculate the time consumption $T_j^1 = t_j^1 \times x_j^1$, which is used by candidate cloud service to complete the task, and store it in the first column of $T_{\text{end}}$. Turn to Step 7. If $1 < n \leq N$, continue, otherwise turn to Step 8.

Step 3: Generate handover matrix $TRANS_{M_{n-1} \times M_n}$, each element of which records the task quantity handed from candidate cloud service of the No. $(n-1)$ subtask module to that of the No. $n$ subtask module. Suppose $m = 1$.

Step 4: If $m \leq M_n$, continue, otherwise turn to Step 7.

Step 5: By traversing the No. $m$ column element of $TRANS_{M_{n-1} \times M_n}$, all the preorder candidate cloud services of candidate cloud service $cs_m^n$ and the number of tasks handed over to $cs_m^n$ are obtained. Then, the start time and task quantity of each task in $cs_m^n$ are obtained, which forms the start state set of $cs_m^n$, $START_m^n = \{(t_1, w_1),(t_2, w_2), \ldots , (t_l, w_l)\}$. Where, $t_l$ is calculated as follows

$$
t_l = \begin{cases} T_l^{n-1} + t_{l \to m}^{n-1} & T_l^{n-1} \neq 0 \\ 0 & T_l^{n-1} = 0 \end{cases} \tag{37}
$$

where, $T_l^{n-1}$ is the end time of the preorder cloud service $cs_l^{n-1}$ of candidate cloud service $cs_m^n$, $t_{l \to m}^{n-1}$ represents the handover time from the preorder cloud service $cs_l^{n-1}$ to the current candidate cloud service $cs_m^n$, and its quantity of handover is $w_l = TRANS_{l \to m}$. If $t_l = 0$, it indicates that there is no task, or no task is handed over to the current cloud service, so $w_l = 0$.

According to the first-handover-first-executed basis, the elements in $START_m^n$ are arranged in ascending order of $t_l$, and $w_l$ is replaced by the end time of the task, that is, $w_l = t_l + t_m^n \cdot w_l$.

The elements in $START_m^n$ are compared in pairs. If $t_{l+1} < w_l$, no operation is performed; if $t_{l+1} < w_l$, $w_{l+1} = w_{l+1} + w_l - t_{l+1}$, $t_{l+1} = w_l$. Therefore, the final $w_{M_{n-1}}$ is the final end time of the cloud service $cs_m^n$, and $w_{M_{n-1}}$ is assigned to the end time matrix $T_{\text{end}}$, that is, $T_m^n = w_{M_{n-1}}$.

Step 6: $m = m + 1$, and turn to Step 4.

Step 7: $n = n + 1$, and turn to Step 7.

Step 8: According to the time consumption matrix $T_{end}$ obtained above, the final time consumption of the cloud service composition scheme $T_{max}$ is the maximum time consumption, with which candidate cloud service of the last subtask module hands over the design results to the resource requester, that is, $T_{\max} = \max\limits_{j \in M_N} \{T_j^N + t_{j \to \text{user}}^N\}$. The process ends.

The pseudo code for calculation of fitness function is shown as Algorithm 6.

---

**Algorithm 6.** The Algorithm of Fitness Function Calculation

---

1. **Input:** experiment time consumption $T_j^1$
handover matrix $TRANS_{M_{n-1} \times M_n}$
2. **Output:** fitness function
3. $T_{end} \leftarrow 0$, $T_{max} \leftarrow 0$, n$\leftarrow$0
4. **if** $(n == 1)$ **then**
5. $T_j^1 \leftarrow t_j^1 \times x_j^1$
6. first row of $T_{end} \leftarrow T_j^1$
7. **go to** 28
8. **else if** $(n \langle 1 \ or \ n \rangle N)$ **then**
9. **go to** 29
10. **end if**
11. Initialization of $TRANS_{M_{n-1} \times M_n}$
12. $m \leftarrow 1$
13. **if** $(m > M_n)$ **then**
14. **go to** 28
15. **end if**
16. $START_m^n \leftarrow \{(t_1, w_1), (t_2, w_2), \ldots, (t_l, w_l)\}$
17. arrange $START_m^n$ in ascending order
18. $w_l \leftarrow t_l + t_m^n \cdot w_l$
19. **for** $(i = 0; i < m - 1; i ++)$ **do**
20. **if** $(t_{i+1} < w_i)$ **then**
21. $w_{i+1} \leftarrow w_{i+1} + w_i \text{-} t_{i+1}$
22. $t_{i+1} \leftarrow w_i$
23. **end if**
24. **end for**
25. $m \leftarrow m + 1$
26. **go to** 13
27. $n \leftarrow n + 1$
28. **go to** 4
29. $T_{max} \leftarrow \underset{j \in M_N}{max} \{T_j^N + t_{j \rightarrow user}^N\}$
30. **return**

---

## 7. Case Study

In order to verify the above proposed method, a case study is carried out in cooperation with an enterprise group. This group is a large enterprise integrating product design and production. It has a number of subsidiaries, and each subsidiary has a relatively independent design department. Due to the imbalance of design capacity, the subsidiaries often need to schedule and match tasks and resources. The group has built a private cloud system, through which the tasks and resources of each subsidiary can interact and match. Based on the above background and the theoretical research proposed in this paper, a case study is carried out. In order to verify the effectiveness of the proposed theory, a multi-level cloud-service matching system based on task hierarchical decomposition is built to complete the matching and intelligent optimization of tasks and resources.

Figure 13 shows the cloud manufacturing prototype system, which mainly provides the functions of task-ontology modeling, task flow decomposition, resource cloud service system, ontology task matching with cloud service system, intelligent optimization of cloud service, and so on. Its structure is shown in Figure 14.

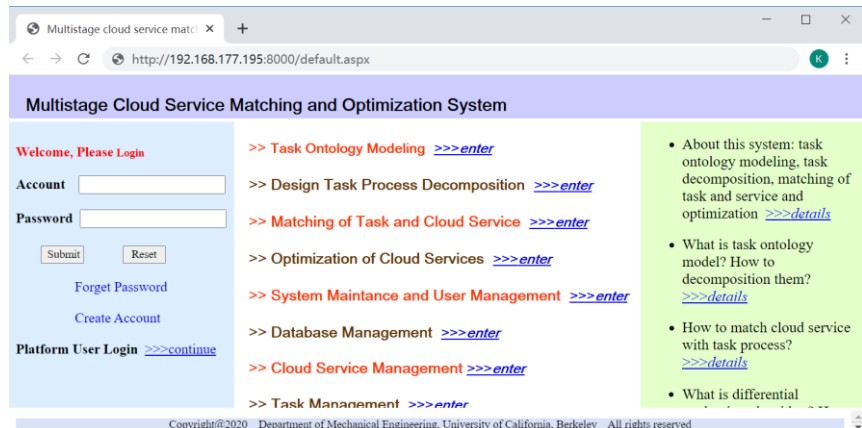

**Figure 13.** Interface of cloud service matching and optimization system.

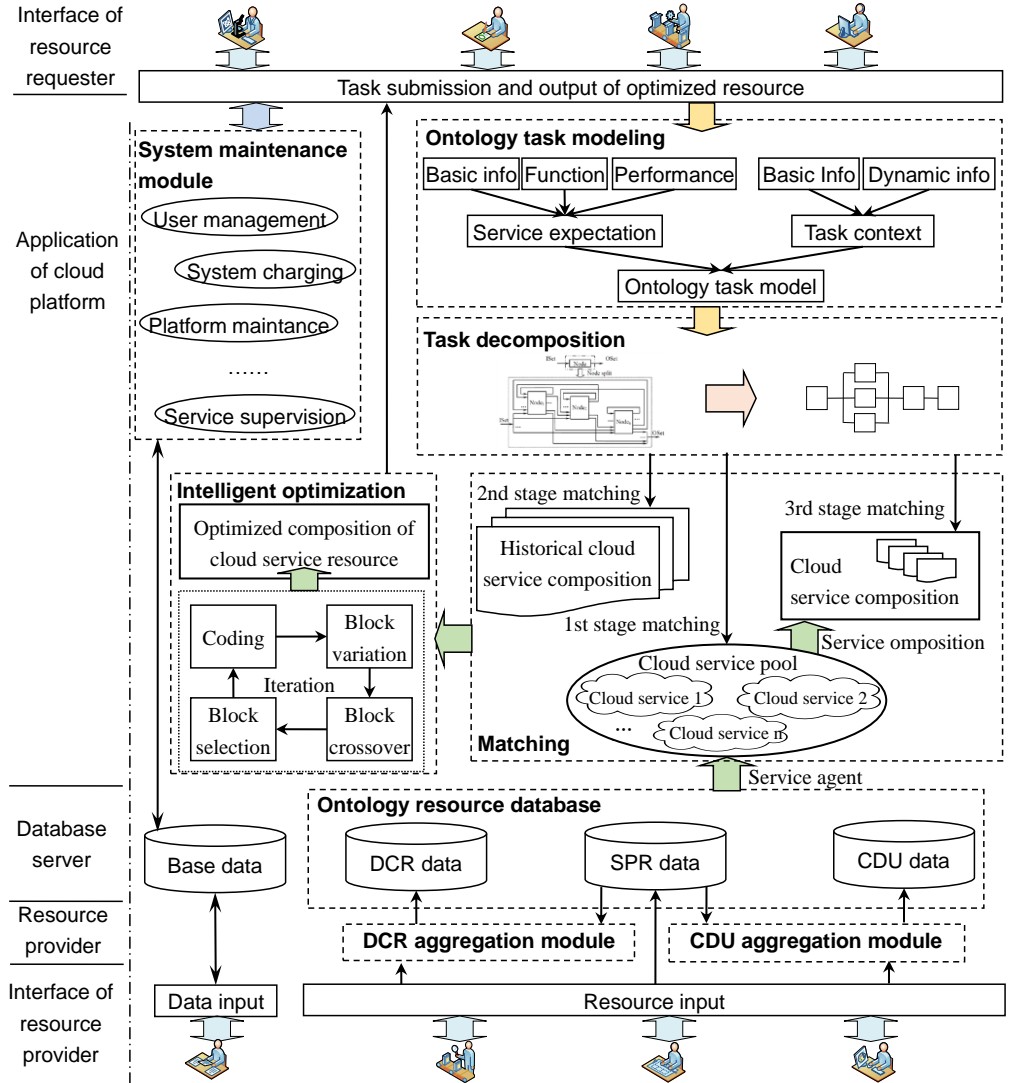

**Figure 14.** Architecture of CMfg system.

The meta resources provided by the resource provider are aggregated to DCRs, which undertakes the design task issued by the resource requester. The aggregation and list of DCRs are shown in Figures 15 and 16. The design task access interface is shown in Figure 17. According to the task granularity, the resource requester set different levels of design tasks.

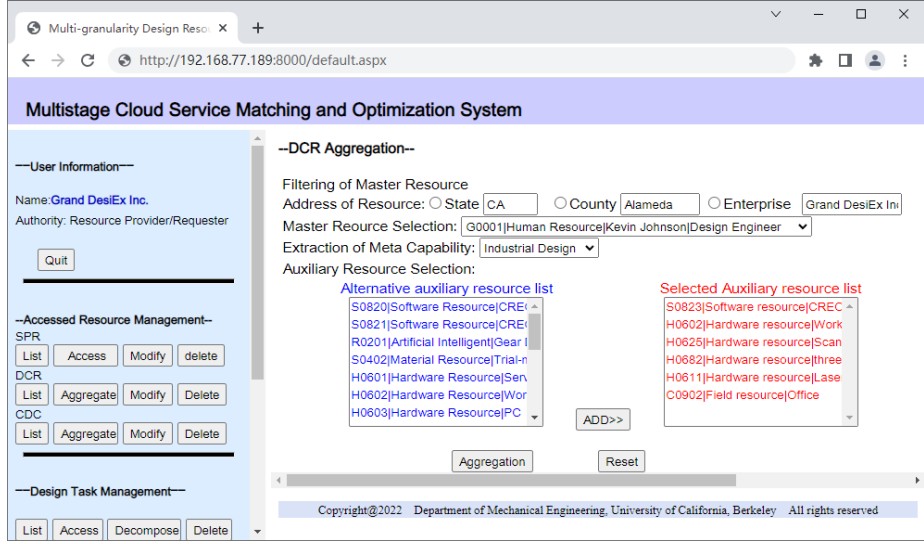

**Figure 15.** Process of DCR aggregation.

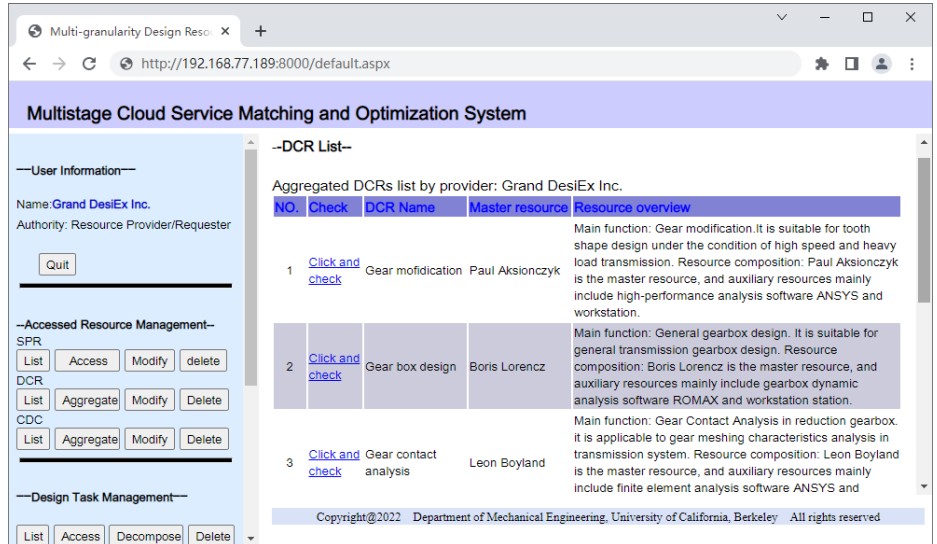

**Figure 16.** DCR list.

For the design of a large mechanical system, a total of 100 special component design tasks are obtained through task modeling and decomposition. Each special component design task includes three sub task modules: overall design, dynamic analysis and detailed design. After matching tasks with cloud services, a number of cloud service resources are found. The consumed time of each task is shown in Table 1, and the handover time consumption between subtasks is shown in Tables 2 and 3.

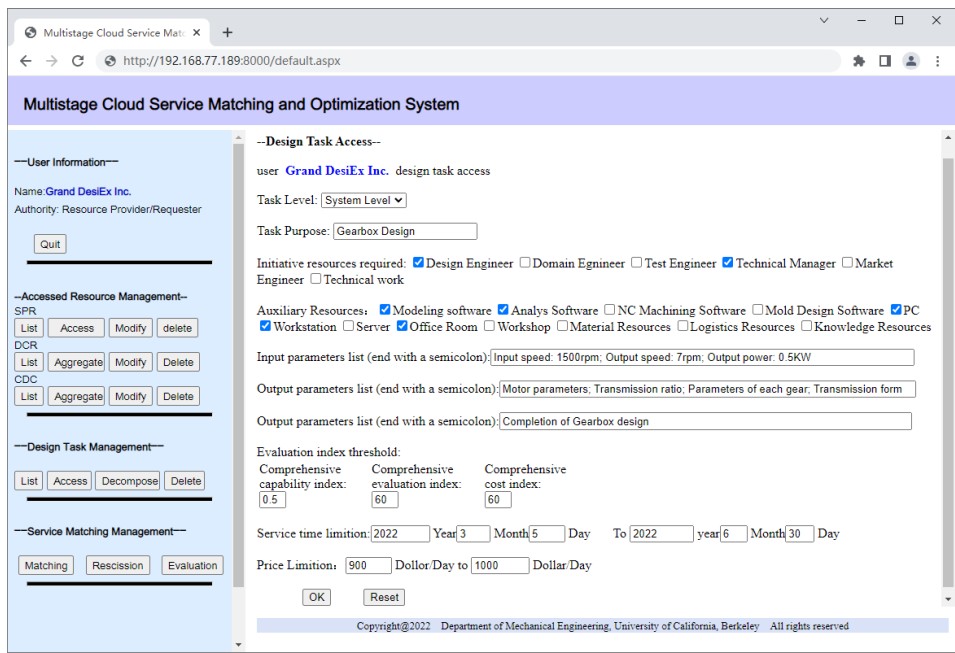

**Figure 17.** The interface of design task access.

**Table 1.** Candidate cloud service module.

| | Subtask 1. | | Subtask 2 | | Subtask 3 |
|---|---|---|---|---|---|
| **Cloud Service** | **Time Consumption (h)** | **Cloud Service** | **Time Consumption (h)** | **Cloud Service** | **Time Consumption (h)** |
| 1 | 50 | 1 | 32 | 1 | 42 |
| 2 | 49 | 2 | 31 | 2 | 41 |
| 3 | 52 | 3 | 30 | 3 | 40 |
| 4 | 51 | 4 | 28 | | |
| 5 | 48 | 5 | 29 | | |
| | | 6 | 33 | | |
| | | 7 | 27 | | |

**Table 2.** The handover time consumption between No.1 and No.2 subtask module (H).

| No. 1 Resource Module | No. 2 Resource Module | | | | | | |
|---|---|---|---|---|---|---|---|
| | **1** | **2** | **3** | **4** | **5** | **6** | **7** |
| 1 | 4 | 5 | 6 | 5.5 | 4.54 | 5.53 | 4.48 |
| 2 | 4.3 | 4.77 | 5.3 | 4.4 | 4.27 | 5.1 | 4.1 |
| 3 | 4.2 | 5.2 | 4.7 | 5.4 | 5.47 | 5.9 | 4.9 |
| 4 | 5.6 | 5.8 | 5.23 | 5.86 | 5.81 | 4.38 | 4.6 |
| 5 | 4.5 | 5.11 | 4.8 | 5.61 | 5.73 | 4.88 | 4.99 |

During the optimization, each chromosome in the population represents a service composition, and the optimization parameters are set as follows: crossover operator CR = 0.5, mutation operator F = 0.5, population size $N_p$ = 30, the maximum number of iterations is 100. The threshold of fitness function is not set and only the maximum evolution algebra is taken as the termination condition of calculation.

**Table 3.** The handover time consumption between No.2 and No.3 subtask module (H).

| No. 2 Resource Module | No. 3 Resource Module | | |
|---|---|---|---|
| | 1 | 2 | 3 |
| 1 | 4 | 5 | 6 |
| 2 | 5.5 | 4.5 | 5.59 |
| 3 | 5.44 | 5.7 | 4.3 |
| 4 | 4.77 | 5.3 | 4.4 |
| 5 | 5.27 | 5.1 | 4.1 |
| 6 | 4.18 | 4.2 | 5.2 |
| 7 | 4.7 | 5.4 | 5.6 |

After completing 100 iterations in MATLAB, the time consumption of each service composition in the population is shown in Table 4, and the objective function curve is shown in Figure 18. In Table 4, the time consumption of No.23 chromosome is the shortest, indicating that in the process of completing the specified design tasks, the time consumption of No.23 service composition is the shortest, and the task allocation of each cloud service resource is shown in Table 5.

**Table 4.** The total time consumption of each chromosome after iteration.

| Chromosomes | 1 | 2 | 3 | 4 | 5 | 6 | 7 | 8 | 9 | 10 |
|---|---|---|---|---|---|---|---|---|---|---|
| Time consumption | 2502.0 | 2523.0 | 2529.8 | 2505.6 | 2513.8 | 2522.0 | 2508.9 | 2550.0 | 2493.0 | 2516.0 |
| Chromosomes | 11 | 12 | 13 | 14 | 15 | 16 | 17 | 18 | 19 | 20 |
| Time consumption | 2534.8 | 2515.0 | 2512.8 | 2510.6 | 2545.0 | 2522.9 | 2529.8 | 2494.0 | 2503.2 | 2512.8 |
| Chromosomes | 21 | 22 | 23 | 24 | 25 | 26 | 27 | 28 | 29 | 30 |
| Time consumption | 2514.0 | 2514.0 | 2490.8 | 2493.0 | 2545.0 | 2508.0 | 2526.0 | 2512.8 | 2508.0 | 2507.8 |

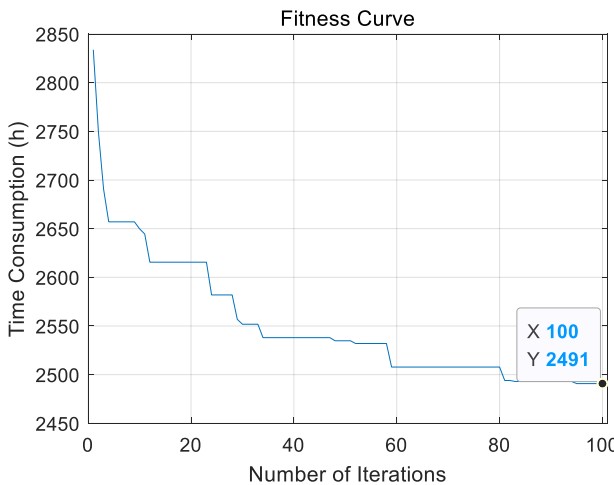

**Figure 18.** Objective function fitness curve.

**Table 5.** The task quantity of each design subtask module.

| Module | Task Quantity of Design Subtask Module | | | | | | | |
|---|---|---|---|---|---|---|---|---|
| No.1 subtask module | Cloud service No. | 1 | 2 | 3 | 4 | 5 | | |
| | Quantity of tasks | 18 | 13 | 16 | 27 | 26 | | |
| No.2 subtask module | Cloud service No. | 1 | 2 | 3 | 4 | 5 | 6 | 7 |
| | Number of tasks | 14 | 17 | 9 | 15 | 15 | 13 | 17 |
| No.3 subtask module | Cloud service No. | 1 | 2 | 3 | | | | |
| | Number of tasks | 33 | 33 | 34 | | | | |

It can be seen from the above example that the improved differential evolution algorithm can further optimize and match the candidate cloud services that meet the threshold requirements and obtain the optimal service composition for a specific design task. After the global search and continuous convergence of the differential evolution algorithm, the optimal task-matching scheme of each cloud service is obtained with the shortest time consumption as the optimization goal.

## 8. Conclusions

In summary, aiming at the matching of tasks and cloud services in a cloud manufacturing system and the optimization of candidate cloud services, a novel multi-level cloud-service matching strategy based on task hierarchical decomposition is proposed, which solves the problem of tasks and cloud services matching with different granularities. A candidate cloud-service optimization algorithm based on improved differential evolution is proposed to intelligently optimize the candidate cloud services and find the optimal service composition with the goal of the shortest time consumption. With the help of a multi-stage cloud-service matching method, the service matching of maximizing task granularity is realized on the premise of ensuring the success rate of matching, which meets the preference of resource requesters for large-granularity service resources. Compared with previous research, our research can realize the optimal matching of design tasks and service resources in the cloud manufacturing environment on the premise of satisfying the preferences of resource users for large-scale service resources. The application platform based on our research can meet the demand of resource requesters for multidisciplinary design resources for interdisciplinary design tasks. Our research realizes the sharing of design resources in the cloud manufacturing environment.

Our future work will focus on the implementation of cloud services to improve the operation and evaluation of cloud services. In addition, the prototype system needs to be improved and expanded to realize the whole process management of tasks and candidate resources.

**Author Contributions:** S.D. has made contributions to the conceptualization, investigation and original draft writing. Z.G. contributed to the methodology, validation and resource. H.W. contributed to methodology, software and validation. F.M. contributed to methodology and draft review & editing. All authors have read and agreed to the published version of the manuscript.

**Funding:** This research was funded by the National Natural Science Foundation of China (Grant No. 52005302).

**Institutional Review Board Statement:** Not applicable.

**Informed Consent Statement:** Not applicable.

**Data Availability Statement:** Not applicable.

**Conflicts of Interest:** The authors declare no conflict of interest.

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
