# Peer review of "Multistage Cloud-Service Matching and Optimization Based on Hierarchical Decomposition of Design Tasks"

_machines, doi:10.3390/machines10090775_

Round 1

Reviewer 1 Report

The authors of this paper aim to show a multistage cloud service matching strategy based on a hierarchical decomposition of design tasks to solve the problem of matching tasks and resources with different granularity sizes. The major issues of this paper are:

[1] The paper requires English proofreading

[2] The abstract is tedious and not concise, and it is suggested to make some modifications to make it more concise.

[3] The authors do not elaborate in the introduction what is the purpose of their review in terms of (i) other similar methodologies, (ii) future research directions, and (iii) identification of research gaps. 

[4] I would like to see a well-developed discussion (minimum of two pages) comparing and contrasting solutions/results presented in the work with existing work and then a subsection of it presenting contributions to theory/knowledge/literature (at least one to two paragraphs) and followed by a subsection on Implications for practice (at least one page). In these paragraphs authors should compare their research approach with previous research, citing references to others' research. 

[5] Authors should reconsider explaining the section about the scientific contribution in the introduction, as well as in the conclusion part of the paper, with a structured comparison of the current research with previous research. The text can be one paragraph long, but it should contain the most important studies.

[6] Please, form the conclusion in the following manner: (i) First paragraph - summary of research and conclusion - e.g. In this paper... ; (ii) Second paragraph - comparison with previous research; (iii) Third paragraph - short description of practical implications; (iv) Fourth paragraph - summary of paper limitations and further implications.

Author Response

Point 1: The paper requires English proofreading

Response 1: We would like to thank the reviewer for your thoughtful review of our manuscript The English writing has been proofread and some improvements have been made, such as lines 39-40, 51-55, 62-65, 77-79, 82-85, 94-95, 104-106, 255-308, 316-318, etc. Please see the revised manuscript.

Point 2: The abstract is tedious and not concise, and it is suggested to make some modifications to make it more concise.

Response 2: Thank you very much for your insightful and constructive advice about the abstract. Some modifications have been made and the abstract is more concise now. Please see the following:

Abstract: In cloud manufacturing system, the multi-granularity of service resource and design task model leads to the complexity of cloud service matching. In order to satisfy the preference of resource requesters for large granularity service resources, we propose a multistage cloud service matching strategy to solve the problem of matching tasks and resources with different granularity sizes. First, a multistage cloud service matching framework is proposed, and the basic strategy of matching tasks with cloud services is planned. Then, the context aware task ontology modeling method is studied, and a context related task ontology model is established. Thirdly, a process decomposition method of design task is studied, and the product development process with small granularity task is established. Fourthly, a matching strategy of ontology task and cloud service is studied, and the preliminary matching is accomplished. Finally, intelligent optimization is carried out, and the optimal cloud service composition is found with the optimal design period as the objective function. With the help of the preceding method, the service matching of maximizing the task granularity is realized on the premise of ensuring the matching success rate, which meets the preference of resource requesters for the large granularity service resources.

Point 3: The authors do not elaborate in the introduction what is the purpose of their review in terms of (i) other similar methodologies, (ii) future research directions, and (iii) identification of research gaps. 

Response 3: We want to thank reviewer for constructive and insightful criticism and advice about the purpose of our review. We addressed it by adding the following: The existing research similar to our study is mainly aimed at the design in non-cloud manufacturing environment, whose application environment is relatively simple. In the future cloud manufacturing environment, the design tasks proposed by resource users will be more complex and the composition of resources will be richer. Under such back-ground, this paper proposes to address the problems of decomposition of coarse granular-ity task, and its multistage matching with cloud services.

Also, it can be seen in lines 51-55 in the revised manuscript.

Point 4: I would like to see a well-developed discussion (minimum of two pages) comparing and contrasting solutions/results presented in the work with existing work and then a subsection of it presenting contributions to theory/knowledge/literature (at least one to two paragraphs) and followed by a subsection on Implications for practice (at least one page). In these paragraphs authors should compare their research approach with previous research, citing references to others' research. 

Response 4: We want to thank reviewer for constructive and insightful criticism and advice about the above question. We updated the manuscript by supplementing some discussion in Sec. 2.5. For the first question, please see the lines 262-302 in the revised manuscript. For the second question, see the lines 303-305. For the third question, see the lines 351-355. 

Point 5: Authors should reconsider explaining the section about the scientific contribution in the introduction, as well as in the conclusion part of the paper, with a structured comparison of the current research with previous research. The text can be one paragraph long, but it should contain the most important studies.

Response 5: Thank you very much for your insightful and constructive advice about the scientific contribution. We updated the manuscript by supplementing lines 62-66 in the introduction, and lines 1152-1153 in the conclusion.

Point 6: Please, form the conclusion in the following manner: (i) First paragraph - summary of research and conclusion - e.g. In this paper... ; (ii) Second paragraph - comparison with previous research; (iii) Third paragraph - short description of practical implications; (iv) Fourth paragraph - summary of paper limitations and further implications.

 Response 6: We want to thank reviewer for constructive and insightful criticism and advice about the conclusion. We updated Sec.8. For (i), we fixed lines 1138-1147. For (ii), we supplemented lines 1147-1150. For (iii), we supplemented lines 1150-1152. For (iv), we fixed lines 1154-1157.

Reviewer 2 Report

The paper presentes  a multistage cloud service matching strategy based on hieararchical decomposition of design tasks to solve the problem of matching tasks and resources with different granularity sizes.

The paper is well written, but is necessary to improve some points of explanation with diagrams.  The desing tasks at the literature review, like the described at lines 95-101 will be more understanble if the authors create a drawing of a diagram to ilustrate the description over these lines.

At section 2.5, the lines 265 to 295 can be improved if the authors create a table that consolidates the discussion, making more clear the pointed issues.

Ate section 5.1 the authors will improve the text if changes the lines 694 to 711 at the algorithm format, as the same showed at algorithm 1 in section 3.

The same applies to lines 924 - 938; 952-968;978-990.

Improves the size of the Figure 11 and Figure 13.

Author Response

Point 1: The paper is well written, but is necessary to improve some points of explanation with diagrams. The desing tasks at the literature review, like the described at lines 95-101 will be more understanble if the authors create a drawing of a diagram to ilustrate the description over these lines.

Response 1: We want to thank reviewer for constructive and insightful advice. We supplemented Figure 1 to ilustrate the description over these lines. Please see Figure 1. at lines 111-112.

Point 2: At section 2.5, the lines 265 to 295 can be improved if the authors create a table that consolidates the discussion, making more clear the pointed issues.

Response 2: Thank you very much for your insightful and constructive advice. We supplemented Figure 2 to clear the description over these lines.

Point 3: Ate section 5.1 the authors will improve the text if changes the lines 694 to 711 at the algorithm format, as the same showed at algorithm 1 in section 3.

Response 3: Thank you very much for your careful review. We supplemented Algorithm 2 to ilustrate these lines.

Point 4: The same applies to lines 924 - 938; 952-968;978-990.

Response 4: Thank you very much for your insightful and constructive advice. We supplemented Algorithm 4, 5 and 6 to ilustrate these lines.

Point 5: Improves the size of the Figure 11 and Figure 13.

Response 5: Thank you very much for your careful review. We had done it. Please see lines 1093 and 1129.

Round 2

Reviewer 1 Report

The authors of the paper improve the original manuscript and must be accepted in the present form